# Formalizing locality for normative synaptic plasticity models

**Colin Bredenberg**[*]
Mila- Quebec AI Institute
Montreal, Quebec H2S 3H1

**Ezekiel Williams**
Mila- Quebec AI Institute
Montreal, Quebec H2S 3H1

**Cristina Savin**
New York University
New York, NY 10003

**Blake Richards**[†]
McGill University
Mila- Quebec AI Institute
Montreal, Quebec H2S 3H1

**Guillaume Lajoie**[†]
Université de Montréal
Mila- Quebec AI Institute
Montreal, Quebec H2S 3H1

## Abstract

In recent years, many researchers have proposed new models for synaptic plasticity in the brain based on principles of machine learning. The central motivation has been the development of learning algorithms that are able to learn difficult tasks while qualifying as "biologically plausible". However, the concept of a biologically plausible learning algorithm is only heuristically defined as an algorithm that is potentially implementable by biological neural networks. Further, claims that neural circuits could implement any given algorithm typically rest on an amorphous concept of "locality" (both in space and time). As a result, it is unclear what many proposed local learning algorithms actually predict biologically, and which of these are consequently good candidates for experimental investigation. Here, we address this lack of clarity by proposing formal and operational definitions of locality. Specifically, we define different classes of locality, each of which makes clear what quantities cannot be included in a learning rule if an algorithm is to qualify as local with respect to a given (biological) constraint. We subsequently use this framework to distill testable predictions from various classes of biologically plausible synaptic plasticity models that are robust to arbitrary choices about neural network architecture. Therefore, our framework can be used to guide claims of biological plausibility and to identify potential means of experimentally falsifying a proposed learning algorithm for the brain.

## 1 Introduction

Over the last several decades, computational neuroscience researchers have proposed a variety of "biologically plausible" models of synaptic plasticity that seek to provide normative accounts of a variety of learning processes in the brain—these models aim to explain how modifications of cellular properties such as excitability and synaptic strengths can improve performance on a variety of important tasks for an organism, using only the information "locally" available to a real neuron in the brain. However, there is no consensus on what "biologically plausible" really means. This is due to a lack of any precise definition of locality, with individual studies adopting different, architecture-specific heuristics that are often not explicitly articulated; further, at an experimental level, the precise quantities available to neurons for computing synaptic updates are still up for debate [1, 2], and likely vary across cell types [3, 4, 5] and brain regions [6, 7, 8, 9, 10].

---

[*]Corresponding author, email at: `colin.bredenberg@mila.quebec`.
[†]Co-senior authors.

37th Conference on Neural Information Processing Systems (NeurIPS 2023).

As an example, some studies postulate that global neuromodulatory signals containing reward information could be available at individual synapses, in addition to the more traditional pre- and post-synaptic information typically required for Hebbian plasticity [11, 12, 1]. Within this broad class, individual studies differ in their assumptions about this reward information (e.g. is it completely global [13] or potentially distinct for different layers or regions [14]) and in the precise details about the pre- and post-synaptic signals available (e.g. is it spike-timing [15] that is available or spike-rate [16]?). Furthermore, there are many more signals that could theoretically be available at synapses: for example, other studies propose that individual pyramidal neurons could receive detailed error (or target) feedback information at their apical dendrites [17, 18, 19, 2] which can be used for learning. Previous work has taxonomized different normative plasticity models according to these feedback signals [20]. Within this space of normative plasticity models, it can be difficult to identify which features of a given model constitute necessary, testable predictions that can be used for experimental verification—i.e., if this information were not available at a synapse, learning with the given algorithm would be impossible—and which features are due to arbitrary choices that may vary with more or less realistic neural network architectures.

As a consequence of these heterogeneous conceptions of locality and given the advancements in the field, it is arguably necessary to develop a clear and precise way to identify what variables synapses should have access to in a learning algorithm, whenever claims of locality are made. In this paper, we focus on formalizing this process. Our central contributions are as follows:

1. We develop an architecture-independent formal framework for locality that we term $\mathbf{S}p$-locality, which requires one to precisely specify the set of variables that synapses are assumed to have access to within a model: different choices of allowed variables for the set $\mathbf{S}$ produce different classes of locality. Importantly, this definition can be used for any learning algorithm applied to any stochastic neural network.

2. We use our framework to group existing plasticity models into different locality classes. Intriguingly, we find that many different algorithms derive their locality from similar principles, even though they belong to different locality classes.

3. We show that our different locality classes make different experimental predictions, which makes it possible to identify which algorithms can be cleanly distinguished experimentally and which cannot.

## 2 Formalizing notions of locality

Our ultimate goal is a formal definition of locality that makes it easy to identify which variables are locally available to synapses in a learning algorithm within any specific model architecture. However, this can be challenging. For example, when we consider a single-compartment neuron model it seems reasonable to assume that the postsynaptic voltage is locally available to synapses. In contrast, in a multi-compartment model, voltages can be different in each compartment, and synapses likely only have access to voltages within nearby compartments. Ideally, we would avoid a definition that simply requires us to list all of the variables that must be made available to the synapse, as that would carry little conceptual weight. However, we will begin by taking this approach, as it will help us to build the groundwork for a more conceptually useful definition in what follows.

**Notation.** We have $\mathbf{X}$ as a random vector in $\mathbb{R}^{N_{\mathbf{X}}}$ denoting network variables (e.g. voltage, spike-rate, inputs, etc.), and $\boldsymbol{\Theta}$ as a random vector in $\mathbb{R}^{N_{\boldsymbol{\Theta}}}$ denoting network parameters (e.g. synaptic weights, spike threshold, etc.). We combine these as $\mathbf{Z} = [\mathbf{X}, \boldsymbol{\Theta}]$—the concatenation of the two vectors. We will use $\mathbf{Z} \sim p(\mathbf{Z})$ to denote a random vector with support $\Omega \subseteq \mathbb{R}^{N_{\boldsymbol{\Theta}}+N_{\mathbf{X}}}$, but will also overload this notation to indicate a vector subject to either universal or existential quantification (e.g. $\exists \mathbf{Z}$ s.t. $\frac{\mathrm{d}f(\mathbf{Z})}{\mathrm{d}\mathbf{Z}} = 0$) or a dummy variable in an expectation (e.g. $\mathbb{E}_{p(\mathbf{Z})}[\mathbf{Z}] = \mu$). We will use the subscript "$\neq i$" to denote removal of the $i^{th}$ element of a vector; for example $\mathbf{Z}_{\neq i} = [\mathbf{Z}_1, \ldots, \mathbf{Z}_{i-1}, \mathbf{Z}_{i+1}, \ldots, \mathbf{Z}_{N_{\mathbf{X}}+N_{\boldsymbol{\Theta}}}]^\top$, and will occasionally use the more general notation $\mathbf{Z}_{\neq \nu}$, where $\nu$ is a set of indices, to denote the vector with the associated set of elements removed.

### 2.1 S-locality: graph-specific locality based on a predefined set of available variables

We begin with a definition of locality that formalizes the concept of a set of variables, $\mathbf{S}_k \subseteq \mathbf{Z}$, to which the $k^{th}$ synapse can have access in order to compute synaptic updates (e.g. some neural voltages,

reward signals, etc.). We then consider the set of these sets across all synapses, $\mathbf{S} = \{\mathbf{S}_1, \ldots, \mathbf{S}_{N_\Theta}\}$. This inspires the following definition:

**Definition 2.1.** *Given parameters* $\Theta \in \mathbb{R}^{N_\Theta}$, *random variables* $\mathbf{X} \sim p(\mathbf{X}|\Theta) \in \mathbb{R}^{N_\mathbf{x}}$, *and function* $f(\cdot) : \mathbb{R}^{N_\mathbf{x}+N_\Theta} \to \mathbb{R}^{N_\Theta}$, *the update function* $f(\mathbf{Z})$ *is* $\mathbf{S}$*-local with respect to some set* $\mathbf{S} = \{\mathbf{S}_k : \mathbf{S}_k \subseteq \mathbf{Z}\}$ *if and only if* $\forall k, i$:

$$\exists \mathbf{Z} \in \Omega \ s.t. \ \frac{\partial f_k(\mathbf{Z})}{\partial \mathbf{Z}_i} \neq 0 \Rightarrow \mathbf{Z}_i \in \mathbf{S}_k. \tag{1}$$

In short, $\mathbf{S}$-locality requires that if parameter update $f_k(\mathbf{Z})$ has a *direct* dependence on $\mathbf{Z}_i$, then $\mathbf{Z}_i$ must be included in the set $\mathbf{S}_k$ of "allowed" variables for that parameter. For example, in the classic approach to locality in a neural network, for any synapse $\mathbf{W}_{ij} \in \Theta$, we might set $\mathbf{S}_{ij} = \{\mathbf{r}_i, \mathbf{r}_j\} \subseteq \mathbf{Z}$, allowing any $\mathbf{S}$-local parameter update under this definition to use pre- ($\mathbf{r}_j$) and post-synaptic ($\mathbf{r}_i$) firing rate information, and nothing else. If we wanted to allow a third factor such as a global neuromodulatory signal, $R$, to project to diffusely to every synapse in the network, as in reward-modulated synaptic plasticity rules, we might take $\mathbf{S}_{ij} = \{\mathbf{r}_i, \mathbf{r}_j, R\}$. In this manner, for any model architecture, we can manually construct a set $\mathbf{S}_{ij}$ that defines our notion of locality for each synapse.

While our definition is intuitive and flexible, it is cumbersome: for every parameter we have to explicitly define which variables are local. This means that $\mathbf{S}$-locality is architecture-specific, requiring an evaluation of which $\mathbf{S}_k$ is an acceptable set of variables for each synapse's update rule. For example, if we switched from a single- to a multi-compartment model, we would now have to modify $\mathbf{S}_{ij}$ to accept only the voltage of the compartment where the synapse is located. Ideally, our definition would make it obvious whether a learning rule is local in both models without having to redefine the list of which variables are local. To do that, we will use the structure of the distribution $p(\mathbf{X}, \Theta)$ to construct a definition of locality that captures intuitions across network architectures.

## 2.2 $p$-locality

Despite its ability to concretely define locality for any model, $\mathbf{S}$-locality is architecture-specific and cumbersome: in this section, we will develop an alternative *operationalized* notion of locality, called $p$-locality (where $p$ stands for "probability"). Our idea is to formulate locality in terms of adjacency in the computational graph underlying network dynamics. To do this we will leverage conditional dependencies in the joint distribution $p(\mathbf{X}, \Theta)$ to generate an architecture-general concept of locality[3]. This is considerably more useful as both a conceptual tool and organizing principle for normative synaptic plasticity models. However, we will still call on $\mathbf{S}$-locality later to formulate our final locality definition: $\mathbf{S}p$-locality.

Further, it is important to note that for this definition, as with $\mathbf{S}$-locality, we do not restrict the functional form of parameter updates $f_k(\cdot)$—we only restrict which variables they are allowed to depend on. This is in contrast to previous, complementary work which has focused on defining allowable local computations and memory complexity constraints for 'biologically plausible' learning algorithms [21]. Our decision not to restrict $f_k(\cdot)$ enables us to abstract away the details of 'allowed' subcellular computations, and will consequently enable us to make much more general statements about the locality properties of normative plasticity algorithms as a whole, without having to focus on the specifics of particular models of neural dynamics.

**Definition 2.2.** *$p$-locality: Consider a probability distribution $p(\mathbf{Z})$ over network activity variables and parameters, and an update function $f : \mathbb{R}^{N_\mathbf{x}+N_\Theta} \to \mathbb{R}^{N_\Theta}$, mapping variables and parameters to parameter updates. Assume that $\frac{\partial p(\mathbf{Z}_i|\mathbf{Z}_{\neq i})}{\partial \Theta_k}$ and $\frac{\partial f_k(\mathbf{Z})}{\partial \mathbf{Z}_i}$ exist for all $\mathbf{Z} \in \Omega$ and $i, k$ indices. The update function $f(\mathbf{Z})$ is local with respect to $p$ ($p$-local) if and only if $\forall k, i$:*

$$\exists \, \mathbf{Z} \in \Omega \text{ s.t. } \frac{\partial f_k(\mathbf{Z})}{\partial \mathbf{Z}_i} \neq 0 \Rightarrow \mathbb{E}_p \left[ \left( \frac{\partial \log p(\mathbf{Z}_i|\mathbf{Z}_{\neq i})}{\partial \Theta_k} \right)^2 \right] \neq 0. \tag{2}$$

To unpack this definition and render it more intuitive, as with $\mathbf{S}$-locality, $\frac{\partial f_k(\mathbf{Z})}{\partial \mathbf{Z}_i}$ indicates a test to see whether the $k$th index of $f$ has a direct dependence on a particular variable $\mathbf{Z}_i$—we assume

---

[3]We define dynamics in terms of probability distributions because many canonical normative plasticity models are stochastic; deterministic computation graphs are a limit case of our approach.

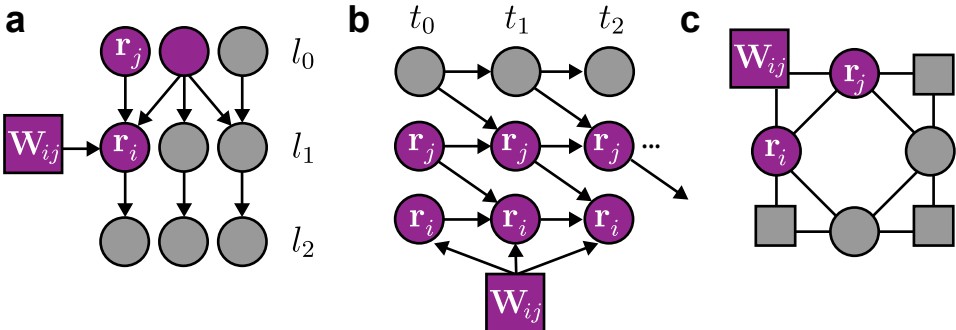

Figure 1: Common neural network graphical models and the permissible parameter updates for a synapse $\mathbf{W}_{ij}$ under $p$-locality. **a.** Feedforward neural networks can be described as DAGs, so variables that are not children or coparents of $\mathbf{W}_{ij}$ are excluded from parameter updates under $p$-locality (by Property 2.1). **b.** For fixed weights in a dynamic neural network like a recurrent neural network (RNN), $p$-locality excludes variables that are not children and coparents of $\mathbf{W}_{ij}$ for all time points (also by Property 2.1). **c.** For an undirected graphical model like a Boltzmann machine, variables that are not neighbors of $\mathbf{W}_{ij}$ are excluded under $p$-locality (by Property 2.2).

that $f_k(\cdot)$ is differentiable primarily for simplicity, and differentiable approximations can always be substituted in practice if $f_k(\cdot)$ happens to not be differentiable. If there is a dependency, then we want that associated random variable $\mathbf{Z}_i$ to be considered local to $\boldsymbol{\Theta}_k$. To measure this, we have selected the Fisher Information of $p(\mathbf{Z}_i|\mathbf{Z}_{\neq i})$ with respect to $\boldsymbol{\Theta}_k$. This measurement quantifies direct influence. For example, if $\mathbf{Z}_i$ is a neuron downstream of neuron $\mathbf{Z}_j$, and $\boldsymbol{\Theta}_k$ is a synapse onto $\mathbf{Z}_j$, then by conditioning on $\mathbf{Z}_j$, $\boldsymbol{\Theta}_k$ is no longer able to have any effect on the conditional probability distribution of $\mathbf{Z}_i$ through its influence on $\mathbf{Z}_j$. In other words, the statistical influence of $\boldsymbol{\Theta}_k$ on $\mathbf{Z}_i$ is mediated *through* the influence of $\mathbf{Z}_j$ on $\mathbf{Z}_i$, and thus it is indirect. By conditioning on $\mathbf{Z}_j$, we seal off one pathway of indirect influence. By conditioning on all $\mathbf{Z}_{\neq i}$, we seal off *all* pathways of indirect influence, which makes $\frac{\partial \log p(\mathbf{Z}_i|\mathbf{Z}_{\neq i})}{\partial \boldsymbol{\Theta}_k} = 0$ for all $\mathbf{Z}$.

It is worth stressing that $p(\mathbf{Z}) = p(\mathbf{X}|\boldsymbol{\Theta})p(\boldsymbol{\Theta})$ is a joint distribution over both $\mathbf{X}$ and $\boldsymbol{\Theta}$, even though most algorithms operate only on the conditional distribution $p(\mathbf{X}|\boldsymbol{\Theta})$. As a consequence, the marginal probability that we place on $p(\boldsymbol{\Theta})$ can be a free choice, which is useful for clarifying the assumptions in the model. For example, we will typically assume that synapses do not have any marginalized dependence on one another, i.e. $p(\boldsymbol{\Theta}) = \prod_k p(\boldsymbol{\Theta}_k)$.

## 2.3 Properties of $p$-locality

Having defined $p$-locality, we can now talk about its interesting properties, which will give intuitions for how it functions and which will figure prominently in our proofs of algorithms' locality. While any $p$-local update could also technically be reformulated as a $\mathbf{S}$-local rule, by taking the set $\mathbf{S}$ to be the set allowed under $p$-locality, the conciseness of $p$-locality and its convenient mathematical properties will prove to be its principal benefits over $\mathbf{S}$-locality. For proofs of these properties, see Appendices A-C. To see the power and generality of $p$-locality as a definition, we first provide two properties that show how one can immediately make statements about the locality of a learning algorithm by simply inspecting graphs associated with the model. Assume all quantities are defined as in Definition 2.2 and that, for certain properties, the joint density satisfies mild regularity constraints (see Appendix C.1). For the first two properties we further assume that $p(\mathbf{Z})$ is strictly positive (see Appendix B). Then we have the following:

**Property 2.1.** *Assume $\mathcal{G}_{\mathrm{d}}$ is a Directed Acyclic Graph (DAG) for $p$. If $\frac{\partial f_k(\mathbf{Z})}{\partial \mathbf{Z}_i} \neq 0$ for $\mathbf{Z}_i$ that is not a parent, co-parent, or child of $\boldsymbol{\Theta}_k$ in $\mathcal{G}_{\mathrm{d}}$, then $f$ is not $p$-local.*

**Property 2.2.** *Assume that $\mathcal{G}$ defines an Undirected Graph (UG) for $p$. If $\frac{\partial f_k(\mathbf{Z})}{\partial \mathbf{Z}_i} \neq 0$ for $\mathbf{Z}_i$ that is not a neighbour of $\boldsymbol{\Theta}_k$ in $\mathcal{G}$, then $f$ is not $p$-local.*

By these properties, as long as the conditional dependencies of $p(\mathbf{Z})$ can be summarized by an UG or DAG (two classes which subsume many modern neural network architectures, see Figure 1 for

examples), we can identify many of the variables disallowed by $p$-locality. Therefore, for such a network, we can decide whether $p$-locality conforms to intuitions about biological plausibility as easily as we can for a hand-crafted set of allowed variables under $\mathbf{S}$-locality; for most practical neural network architectures, we will see below that $p$-locality does behave intuitively. Thus, with $p$-locality we have a definition of locality that does not require an exhaustive list of the variables available to a synapse, but rather, which relies on the implicit computational structure of the model (i.e. $p(\mathbf{Z})$).

**Property 2.3.** *For any function $b : \mathbb{R}^{N_{\mathbf{X}}+N_{\Theta}} \rightarrow \mathbb{R}^{N_{\Theta}}$ defined such that $b_k(\mathbf{Z}) = h_k(f_k(\mathbf{Z}), g_k(\mathbf{Z}))$, where $f(\mathbf{Z})$ and $g(\mathbf{Z})$ are $p$-local and $h_k$ is differentiable, $b(\mathbf{Z})$ is also $p$-local.*

This demonstrates that $p$-local functions can be arbitrarily combined without the combination losing the $p$-local property. It also shows that $p$-locality places no restrictions on the functional form of parameter updates, so long as they are exclusively functions of variables allowed under $p$-locality.

**Property 2.4.** *For any function $f(\cdot) : \mathbb{R}^{N_{\mathbf{X}}+N_{\Theta}} \rightarrow \mathbb{R}^{N_{\Theta}}$, there exists a probability distribution $p(\mathbf{Z})$ such that the random variable $f(\mathbf{Z})$ with $\mathbf{Z} \sim p(\mathbf{Z})$ is $p$-local.*

This is intended as a cautionary note: unless $p$ defines a neural network-like probability distribution, $p$-locality does not necessarily conform to intuitions about biological locality (see Appendix F for examples). That $f(\mathbf{Z})$ is $p$-local for *some* probability distribution says nothing about a function $f(\mathbf{Z})$; $p$-locality only becomes informative when we narrow our focus to probability distributions with biological relevance.

The following properties will turn out to be extremely important in Section 3: all algorithms that we survey below with provable $p$-locality properties will involve the score function or the derivative of the unnormalized energy function.

**Property 2.5.** *The derivative of the log joint distribution $\frac{\partial \log p(\mathbf{X}, \Theta)}{\partial \Theta}$ is $p$-local.*

**Property 2.6.** *For a probability distribution given by $p(\mathbf{Z}) = \frac{1}{\mathcal{Z}} \exp\left(-E(\mathbf{Z})\right)$, where $\mathcal{Z}$ is a normalizing constant, the expression $\frac{\partial}{\partial \Theta} E(\mathbf{Z})$ is $p$-local.*

**Property 2.7.** *If the parameter marginal distribution factorizes as $p(\Theta) = \prod_k p(\Theta_k)$, i.e. the parameters are independent from one another, then the score function $\frac{\partial \log p(\mathbf{X}|\Theta)}{\partial \Theta}$ is $p$-local.*

**Property 2.8.** *For a mixture distribution $p_{12}(\mathbf{Z}, \gamma) = p_1(\mathbf{Z})^\gamma p_2(\mathbf{Z})^{1-\gamma} p(\gamma)$ for some binary variable $\gamma \in \{0, 1\}$ with nonzero probabilities, if $f(\mathbf{Z})$ is $p_1$-local (or equivalently $p_2$-local), then $f(\mathbf{Z})$ is $p_{12}$-local.*

Intuitively, this feature holds because mixture distributions introduce *more* dependencies between parameters $\Theta$ and random variables $\mathbf{X}$. Therefore, $p$-locality under a mixture of probability distributions is more permissive than $p$-locality under any single one of its constituent probability distributions.

## 2.4 A simple motivating example

To make our definition of $p$-locality more concrete, we will now explore how it functions for a common network architecture. Consider a simplified linear-nonlinear feedforward neuron model with $L$ layers of neurons $\mathbf{r}^{(l)}$ ($\mathbf{X} = \left[\mathbf{r}^{(1)}, ..., \mathbf{r}^{(L)}\right]$) connected by synaptic weights $\mathbf{W}^{(l)}$ ($\Theta = \left[\mathbf{W}^1, ..., \mathbf{W}^{(L)}\right]$), with additive Gaussian noise at each layer. Conditioned on our feedforward weight matrices and stimuli $\mathbf{s}$, the probability distribution of neural firing rates is given by:

$$p(\mathbf{r}|\mathbf{s}, \mathbf{W}) = \prod_{l=1}^{L} \prod_{i=1}^{N_l} p(\mathbf{r}_i^{(l)}|\mathbf{r}^{(l-1)}, \mathbf{W}^{(l)}) \tag{3}$$

$$p(\mathbf{r}_i^{(l)}|\mathbf{r}^{(l-1)}, \mathbf{W}^{(l)}) \sim \mathcal{N}(h(\mathbf{W}_{i:}^{(l)} \mathbf{r}^{(l-1)}), \sigma^2), \tag{4}$$

where for notational simplicity we have taken $\mathbf{r}^{(0)} = \mathbf{s}$, $h(\cdot)$ is a pointwise nonlinearity, and $\mathbf{W}_{i:}^{(l)}$ corresponds to the $i$th row of $\mathbf{W}^{(l)}$. As discussed above, most algorithms operate on the conditional distribution, $p(\mathbf{X}|\Theta)$, so we only define a prior over parameters for the purposes of assessing $p$-locality. To ensure that our joint distribution is a directed graphical model, and to ensure that we have no dependencies between parameters, we will assume that the parameters for the network are all independently distributed, i.e. $p(\mathbf{W}) = \prod_{i,j,l} p(\mathbf{W}_{ij}^{(l)})$, where $\mathbf{W}_{ij}^{(l)}$ is the synapse between

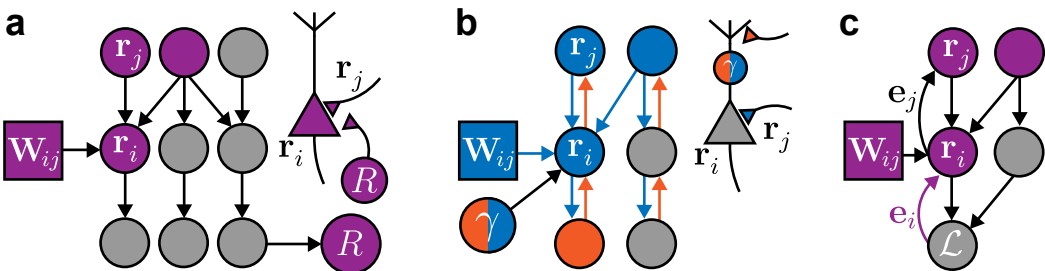

Figure 2: Different versions of $\mathbf{S}p$-locality. **a.** $Rp$-locality functions as $p$-locality, but also allows synapses to use rewards $R$ for parameter updates, corresponding to an additional diffuse neuromodulatory signal. **b.** $\gamma p_{md}$-locality allows synapses to use $p_m$-local variables *and* $p_d$-local variables, as well as the global gating variable $\gamma$. For the Wake-Sleep algorithm, a $\gamma p_{md}$-local probability distribution could correspond to a network of pyramidal neurons receiving $p_m$-related synapses in their apical compartments (orange) and $p_d$-related synapses in their basal compartments. The variable $\gamma$ controls whether $p_d$ or $p_m$ synapses affect cellular dynamics. **c.** Backpropagation and its approximations are $\mathbf{e}_i p$-local, which functions like $p$-locality, except it additionally permits a neuron-specific error signal to be used for parameter updates. Typically, $\mathbf{e}$ is constructed by sequentially propagating error signals backwards through a feedforward network.

postsynaptic neuron $i$ and presynaptic neuron $j$ in layer $l$. To make our choice of conditional probability distribution more concrete (see Figure 1a), we can see that neural firing rates for this model can be sampled by:

$$\mathbf{r}^{(l)} = h(\mathbf{W}^{(l)}\mathbf{r}^{(l-1)}) + \sigma\boldsymbol{\eta}^{(l)}, \tag{5}$$

where $\boldsymbol{\eta}^{(l)} \sim \mathcal{N}(0, 1)$. This corresponds to an ordinary multilayer perceptron neural network with noise added at every layer. We can use this model to get intuition for how this choice of probability distribution $p$ constrains the variables allowed for an update for a particular parameter $\mathbf{W}_{ij}^{(l)}$.

Now, our probability distribution corresponds to a directed graphical model with dependencies given by Eq. 3, so by Property 2.1, we know that variables that are not parents, coparents, or children of $\mathbf{W}_{ij}^{(l)}$ violate $p$-locality if included in the parameter update for $\mathbf{W}_{ij}^{(l)}$. Figure 1 summarizes the variables allowed under $p$-locality for such a DAG. $\mathbf{W}_{ij}^{(l)}$ has no parents under our graphical model, leaving only children and coparents. $\mathbf{W}_{ij}^{(l)}$ has only one child, $\mathbf{r}_i^{(l)}$, whose coparents are given by $\mathbf{W}_{ik}^{(l)}$ s.t. $k \neq j$ (all other synapses onto $\mathbf{r}_i^{(l)}$), and $\mathbf{r}^{(l-1)}$ (all presynaptic neurons).

This shows that updates for synapses in our simple feedforward network cannot depend on neurons that are in layers $> l$ or $< l - 1$ while maintaining $p$-locality. We also found that dependencies on neurons within the postsynaptic layer $l$ for indices $k \neq i$ were also not permissible. As a concrete example, by Property 2.7, we found that the score function of $p(\mathbf{X}|\boldsymbol{\Theta})$ is $p$-local if the parameters are assumed to be independently distributed. Our example satisfies this criterion. Its score function is given by: $\frac{\partial \log p(\mathbf{r}|\mathbf{s},\mathbf{W})}{\partial \mathbf{W}_{ij}^{(l)}} = \frac{\left(\mathbf{r}_i^{(l)} - h(\mathbf{V}_i^{(l)})\right)}{\sigma^2} h'(\mathbf{V}_i^{(l)})\mathbf{r}_j^{(l-1)}$, where $\mathbf{V}_i^{(l)} = \mathbf{W}_{i:}^{(l)}\mathbf{r}^{(l-1)}$. By inspection, this function only requires information about the post- and presynaptic neurons, as well as the summed input to the postsynaptic neuron, $\mathbf{V}_i$.

This simple example clearly outlines why $p$-locality aligns with expected notions of locality. More importantly, the use of $p$-locality also has the benefit of generalizing cleanly to alternative network architectures. For example, if we were to instead inspect a multilayer recurrent neural network (Figure 1b), we would see that for $\mathbf{W}_{ij}^{(l)}$ the firing rates $\mathbf{r}_i^{(l)}(t)$ and all presynaptic neural firing rates would be permissible *for all time steps*, because the activities at each timestep are children or coparents of $\mathbf{W}_{ij}^{(l)}$. For another classical example of a biologically plausible probability distribution corresponding to an UG (as in Figure 1c.), see Appendix D for a discussion of a linear continuous Boltzmann machine.

## 2.5 S$p$-locality

In almost all situations, a given class of plasticity algorithms will produce updates that are *almost* $p$-local, in that synapses are required to have access to some small number of global state variables that a biological system could in principle project diffusely throughout a network, but which would not be allowed under generic $p$-locality (e.g. reward, which is usually produced far downstream of any given synapse in a neural network). For this eventuality, we introduce a notion called S$p$-locality, which is best considered a hybrid of S-locality and $p$-locality.

**Definition 2.3.** *Given parameters $\mathbf{\Theta} \in \mathbb{R}^{N_\Theta}$ and random variables $\mathbf{X} \in \mathbb{R}^{N_\mathbf{x}}$, concatenated as $\mathbf{Z} = [\mathbf{X}, \mathbf{\Theta}]^\top$, and a function $f(\mathbf{Z}) : \mathbb{R}^{N_\mathbf{x}+N_\Theta} \to \mathbb{R}^{N_\Theta}$, the update function $f(\mathbf{Z})$ is S$p$-local with respect to probability distribution $p(\mathbf{Z})$ and some set $\mathbf{S} = \{\mathbf{S}_k : \mathbf{S}_k \subseteq \mathbf{Z}\}$ if and only if $\forall k$ $f_k = h_k(\mathbf{S}_k, g_k(\mathbf{Z}))$, where $h_k$ is an arbitrary function and $g_k(\mathbf{Z})$ is $p$-local.*

The value of this definition is that it negotiates a compromise between the architecture-generality of $p$-locality, and the flexibility of S-locality. Obviously if we take $\mathbf{S}_k = \mathbf{Z}$ $\forall k$, all functions are S-local, and likewise, if we define $p$ sufficiently generally, all functions are $p$-local (Property 2.4). However, if some class of optimization algorithms operating on a predefined probability distribution $p$ are provably guaranteed to produce parameter updates that are S$p$-local for some small set S, then we will have obtained a very concise description of the types of information required by that algorithm.

## 3 Applying S$p$-locality to normative plasticity models

Given a probability distribution over a neural network states and parameters, $p(\mathbf{X}, \mathbf{\Theta})$, which could be experimentally motivated or even observed, it is natural to ask which learning algorithms respect locality principles in the way parameter updates are made during learning. Alternatively, if we denote an algorithm which operates on a probability distribution $p(\mathbf{X}, \mathbf{\Theta})$ to output an update $f(\mathbf{Z})$ by $\mathcal{A}(p(\mathbf{X}, \mathbf{\Theta})) = f(\mathbf{Z})$, where $f(\mathbf{Z}) : \mathbb{R}^{N_\mathbf{x}+N_\Theta} \to \mathbb{R}^{N_\Theta}$, we can ask which probability distributions $p(\mathbf{X}, \mathbf{\Theta})$ and variable collections S leave $f(\mathbf{Z})$ S$p$-local. This is one way S$p$-locality could potentially facilitate experimentally testing biologically plausible learning models.

To demonstrate the utility of our approach, in this section we characterize the S$p$-locality properties of a wide variety of algorithms that have historically been used to produce normative models of synaptic plasticity, as summarized in Table 1. Though our framework covers many algorithms, here we expand on three prototypical algorithms in detail, focusing on features that are relevant for our discussion of the algorithms' locality properties. More algorithmic details as well as theorem proofs are provided in Appendix E. Schematics depicting biological interpretations for the types of S$p$-locality discussed below are provided in Figure 2.

### 3.1 REINFORCE

REINFORCE [13], also known as policy gradient learning, produces reward-modulated Hebbian parameter updates for neural networks [1] similar to the one discussed in Section 2.4. Here we will show that REINFORCE is $Rp$-local, meaning that it is S$p$-local where $\mathbf{S} = \{\mathbf{S}_k = R \;\; \forall k\}$, which assumes each synapse has access to a global scalar reward signal $R$ that needs to be delivered diffusely to all synapses in a network.

**Theorem 3.1.** *If $p(\mathbf{\Theta}) = \prod_k p(\mathbf{\Theta}_k)$, the REINFORCE estimator given by $\mathcal{A}_R(p(R, \mathbf{X}|\mathbf{\Theta}))$ is $Rp$-local.*

As an example, consider the network defined by Eq. 3. The REINFORCE update for this network, for a single sample of the network state $\mathbf{r}$ and reward $R$, is given by:

$$\Delta \mathbf{W}_{ij}^{(l)} = R \frac{\partial \log p(\mathbf{r}|\mathbf{s}, \mathbf{W})}{\partial \mathbf{W}_{ij}^{(l)}} = R \frac{\left(\mathbf{r}_i^{(l)} - h(\mathbf{V}_i^{(l)})\right)}{\sigma^2} h'(\mathbf{V}_i^{(l)})\mathbf{r}_j^{(l-1)}, \tag{6}$$

where again $\mathbf{V}_i^{(l)} = \mathbf{W}_{i:}^{(l)}\mathbf{r}^{(l-1)}$. This is just the scalar multiplication of the reward signal $R$ with the score function; because the score function is $p$-local, the full update is $Rp$-local (see Appendix E for more detail).

## 3.2 Wake-Sleep (WS)

Whereas REINFORCE has been used to model reinforcement learning in neural networks, the Wake-Sleep algorithm has been used to model unsupervised learning [22]. It assumes that a neural network has two modes of operation, controlled by a scalar variable $\gamma$ that determines whether the network is in inference mode ('wake', $\gamma = 1$) or in generative mode ('sleep,' $\gamma = 0$). In all, neural activity samples are drawn from the mixture distribution given by $p_{md}(\mathbf{r}, \gamma | \mathbf{W}, \mathbf{M}) = p_d(\mathbf{r}|\mathbf{W})^{\gamma} p_m(\mathbf{r}|\mathbf{M})^{(1-\gamma)} p(\gamma)$, where $p(\gamma)$ defines the probability of sampling from the wake phase ($p_d$; $d$ corresponds to 'data') or the sleep phase ($p_m$; $m$ corresponds to 'model'), and $\mathbf{W}$ corresponds to the feedforward weights while $\mathbf{M}$ correspond to feedback weights. For conciseness, in all subsequent sections we will denote the mixture distribution $p_1(\mathbf{Z})^{\gamma} p_2(\mathbf{Z})^{1-\gamma} p(\gamma)$ by $Mix\,(p_1(\mathbf{Z}), p_2(\mathbf{Z}))$. As an example, building on the feedforward network defined by Eq. 3 by adding a generative feedback pathway, we have neural dynamics given by:

$$\mathbf{r}^{(l)} = \gamma h_w(\mathbf{W}^{(l)}\mathbf{r}^{(l-1)}) + (1-\gamma)h_s(\mathbf{M}^{(l)}\mathbf{r}^{(l+1)}) + \sigma\boldsymbol{\eta}^{(l)}, \tag{7}$$

where $h_w(\cdot)$ corresponds to the 'wake' nonlinearity, and $h_s(\cdot)$ corresponds to the 'sleep' nonlinearity. Under this formulation, $\gamma$ gates, for *all neurons*, whether activity is driven by the feedforward or feedback pathways; there are several hypotheses for how this could be implemented at a neuronal level, including global neuromodulatory or inhibitory gating of the apical and basal dendrites of pyramidal neurons in the cortex [23] (Fig 2b).

The Wake-Sleep algorithm updates $\mathbf{M}$ to fit $p_m$ as a generative model of the network's input data, and updates $\mathbf{W}$ to fit $p_d$ to perform approximate inference with respect to that generative model, in a manner similar to variational autoencoders (VAE) [24, 25]; unlike VAE training with backpropagation, however, the Wake-Sleep algorithm is $\gamma p_{md}$-local, meaning $\mathbf{S} = \{S_k = \gamma \ \forall k\}$ (see Appendix E for the proof).

**Theorem 3.2.** *If* $p(\boldsymbol{\Theta}, \boldsymbol{\Theta}^{(d)}) = \left(\prod_k p(\boldsymbol{\Theta}_k)\right)\left(\prod_k p(\boldsymbol{\Theta}_k^{(d)})\right)$, *the Wake-Sleep estimator given by* $\mathcal{A}_{WS}(p_m(\mathbf{X}|\boldsymbol{\Theta}), p_d(\mathbf{X}|\boldsymbol{\Theta}^{(d)}))$ *is* $\gamma p_{md}$-*local, where* $p_{md} = Mix\,\left(p_m(\mathbf{X}|\boldsymbol{\Theta}), p_d(\mathbf{X}|\boldsymbol{\Theta}^{(d)})\right) p(\boldsymbol{\Theta}, \boldsymbol{\Theta}^{(d)})$.

As a concrete example, the Wake-Sleep parameter update for $\mathbf{W}_{ij}^{(l)}$ for a single sample of the state $\mathbf{r}$ from the network above is given by:

$$\Delta\mathbf{W}_{ij}^{(l)} = (1-\gamma)\frac{\partial \log p_d(\mathbf{r}|\mathbf{s}, \mathbf{W})}{\partial \mathbf{W}_{ij}^{(l)}} = (1-\gamma)\frac{\left(\mathbf{r}_i^{(l)} - h_w(\mathbf{V}_i^{(l)})\right)}{\sigma^2}h_w'(\mathbf{V}_i^{(l)})\mathbf{r}_j^{(l-1)}, \tag{8}$$

where $\mathbf{V}_i^{(l)} = \mathbf{W}_{i:}^{(l)}\mathbf{r}^{(l-1)}$; a similar update holds for $\mathbf{M}$. Similar to REINFORCE, this parameter update is only the combination of a scalar variable $\gamma$ and the score function, and is consequently $\gamma p_{md}$-local (see Appendix E for the proof). Interestingly, Wake-Sleep is not the only algorithm to obey this form of locality: several other normative plasticity models, including Boltzmann machine learning [26], equilibrium propagation [27], and impression learning [23] have essentially the same $p$-locality properties. However, some of these have additional requirements on the probability distribution that make them less biologically plausible. For example, Boltzmann machine learning is $\gamma p_{md}$-local only for distributions that can be captured with energy-based models (which typically require weight symmetry).

## 3.3 Backpropagation (BP) and its approximations

Here we provide a characterization of the locality of the backpropagation algorithm [28], which as we will show using our definitions, is only a local algorithm with biologically implausible assumptions [20]. This demonstration is primarily important because the $\mathbf{S}p$-locality properties of several biologically plausible backpropagation approximations, including feedback alignment [29], weight mirror [30], and Burstprop [31] satisfy a similar notion of $\mathbf{S}p$-locality, but using more biologically realistic assumptions about $\mathbf{S}$ (see Table 1 and Appendix E). Derivations for the backpropagation algorithm and its approximations require more stringent network assumptions (here we assume a feedforward multilayer perceptron network as in Section 2.4), because the algorithm itself is not well-defined for arbitrary probabilistic network architectures.

| Optimization Alg. | Locality supported | Architectural restrictions |
|---|---|---|
| REINFORCE | $Rp$-local | DAG |
| Maximum Likelihood Estimation | $p_m$-local | DAG |
| Generalized EM | $p_m$-local | DAG |
| Predictive Coding | $p_m$-local | MAP (weight symmetry) |
| Wake-Sleep | $\gamma p_{md}$-local | DAG |
| Impression learning | $\gamma p_{md}$-local | DAG |
| Contrastive Divergence | $\gamma p_{md}$-local | K-Step EBM (weight symmetry) |
| Equilibrium Propagation | $\gamma p_{md}$-local | EBM (weight symmetry) |
| WTA-STDP | $p_m$-local | WTA |
| Backpropagation | $\mathbf{e}_i^{(l)} p$-local | MLP (weight transport) |
| Feedback alignment | $\hat{\mathbf{e}}_i^{(l)} p$-local | MLP |
| Weight mirror | $\hat{\mathbf{e}}_i^{(l)} p$-local | MLP |
| Burstprop | $\hat{\mathbf{e}}_i^{(l)} p$-local | MLP |
| RTRL | $\mathbf{eJ}p$-local | RNN |
| e-prop | $\mathbf{e}_i p$-local | RNN |
| RFLO | $\hat{\mathbf{e}}_i p$-local | RNN |
| FOLLOW | $p$-local | error-RNN |

Table 1: **Summarizing S$p$-locality for classical learning algorithms.** First column: the optimization algorithm of interest. Second column: the variant of **S**$p$-locality that we have proven in Appendix E. Third column: the network architectures required for both the algorithm and our proofs to work. DAG = Directed Acyclic graphical model, MAP = Maximum *a posteriori* gradient descent dynamics, WTA = winner-take-all circuit, EBM = Energy-based model, MLP = Multilayer perceptron, RNN = recurrent neural network, error-RNN = error-driven recurrent neural network. 'K-Step' refers to the small, finite number of sampling steps required, as opposed to sampling from a steady-state distribution or calculating an equilibrium, which are more time-intensive. Weight symmetry indicates that symmetric recurrent connectivity is required for the original proposed algorithm. Weight transport indicates that symmetric weights are required exclusively for error propagation.

**Theorem 3.3.** *If* $p(\mathbf{\Theta}) = \prod_k p(\mathbf{\Theta}_k)$ *and* $p(\mathbf{X}|\mathbf{\Theta})$ *is defined by Eq. 3, the BP update for* $\mathbf{W}_{ij}^{(l)}$ *with a loss* $\mathcal{L}(\mathbf{X})$*, given by* $\mathcal{A}_{BP}(p(\mathbf{X}|\mathbf{\Theta}), \mathcal{L}(\mathbf{X}))$ *is* $\mathbf{e}_i^{(l)} p$*-local, where* $\mathbf{e}_i^{(l)} = \frac{\mathrm{d}L}{\mathrm{d}\bar{\mathbf{r}}_i}$*. Similarly, the updates for feedback alignment, weight mirror, and Burstprop are* $\hat{\mathbf{e}}_i^{(l)} p$*-local, where* $\hat{\mathbf{e}}_i^{(l)}$ *is given by their respective gradient approximations.*

This notion of $\hat{\mathbf{e}}_i^{(l)} p$-locality, where $\mathbf{S} = \{\mathbf{S}_{ij}^{(l)} = \hat{\mathbf{e}}_i^{(l)} \ \ \forall i, j, l\}$, includes more restrictive biological assumptions than the one provided by $Rp$-locality and $\gamma p_{md}$-locality, because it relies heavily on the neuron-specific error $\hat{\mathbf{e}}_i^{(l)}$, which must be then accounted for using physiological mechanisms. For our network architecture given by Eq. 3, the functional form of all backpropagation approximations listed in Table 1 is given by:

$$\Delta \mathbf{W}_{ij}^{(l)} = -\hat{\mathbf{e}}_i h'(\mathbf{V}_i^{(l)})\mathbf{r}_j^{(l-1)}, \tag{9}$$

where $\hat{\mathbf{e}}_i$ is an algorithm-specific approximation of $\frac{\mathrm{d}L}{\mathrm{d}\bar{\mathbf{r}}_i}$. Similar to $\gamma p_{md}$-local algorithms, recent work has suggested that this error signal could be propagated through networks by a top-down error signal relayed to the apical dendrites of pyramidal neurons [31] (see Fig. 2c). Thus, these algorithms make specific biological predictions, driven by the need for $\hat{\mathbf{e}}_i^{(l)}$ to be accessible to each synapse.

## 4 Discussion

We have constructed a formal definition of locality, **S**$p$-locality, that combines set-based and probabilistic graphical model-based formalisms (**S**-locality and $p$-locality, respectively). Our goal was standardizing and operationalizing notions of locality, and we also demonstrated the usefulness of these definitions in characterizing normative plasticity models. Notably, this approach enabled the identification of distinct classes of **S**$p$-locality (Table 1) which subsume learning algorithms spanning

more than thirty years of research, from Boltzmann machines [26] to modern incarnations like equilibrium propagation [27]. Importantly, our framework can be applied to any neural network architecture, incorporating networks that are recurrent [16, 32, 23, 27, 26, 33], spiking [34, 12], and multilayer [29, 22, 30] into a single framework. This is thanks, in large part, to the architecture-generality of $p$-locality, which abstracts away specific details of network models, defining locality instead in terms of statistical dependencies—this makes our framework much more powerful than simpler linguistic descriptions of locality. As we discuss in Appendix F, this abstraction comes at the cost of requiring researchers to specify a distribution, $p$, that is grounded in biological constraints. We note that for non-biological probability distributions the variables allowed to be "local" under $p$-locality can violate standard intuitions; however, for standard neural networks like those discussed in Section 2.4 and Appendix D, $p$-locality behaves as one would expect, with "local" learning rules being those that use pre- and postsynaptic activity.

Table 1 organizes existing normative plasticity models into distinct classes. The first locality class, $Rp$-locality, requires individual synapses to have access to reward information; this class encompasses many reward-modulated Hebbian plasticity models [1]. The second class, $p_{(m)}$-locality, encompasses predictive coding and its variations [33, 35], and aligns with more traditionally Hebbian plasticity updates (often at the cost of unrealistic network architectures; see Table 1). The third class, $\gamma p_{md}$-locality, requires neural networks to have two distinct modes of operation (e.g. the 'wake' and 'sleep' modes in the Wake-Sleep algorithm) gated by a scalar global variable $\gamma$. Thus, $\gamma p_{md}$-locality requires synapses to have access to the value of $\gamma$ to compute updates, and subsumes many algorithms that require generative feedback pathways for learning [22, 23]. The final class of algorithms are $\hat{\mathbf{e}}_i^{(l)} p$-local for feedforward neural networks, where $\hat{\mathbf{e}}_i^{(l)}$ is a neuron-specific approximate error signal, and includes feedback alignment [29], weight mirror [30], and Burstprop [31], as well as temporal variants of these algorithms [36, 37]. These four categories—$Rp$-locality, $p_{(m)}$-locality, $\gamma p_{md}$-locality, and $\hat{\mathbf{e}}_i^{(l)}$-locality—subsume, to our knowledge, almost all existing normative plasticity models.

Because these plasticity models are sometimes only constructed for specific network architectures (which may violate known facts about the brain), we also specify the broadest class of network architectures for which we have successfully derived a model's $\mathbf{S}p$-locality properties (proven in Appendix E). As noted in Appendix F, 'biological plausibility' requires both network architectures and parameter updates to be in line with existing experimental evidence, while $\mathbf{S}p$-locality provides a formal framework for assessing only the latter of these two factors. We therefore stress that $\mathbf{S}p$-locality properties are only relevant for neuroscience if they can be determined for biologically plausible network architectures. As a consequence, classes of network architectures that encompass a broad range of biologically plausible network models (such as DAGs) are desirable, whereas algorithms that are only viable for implausible architectures (e.g. those that imply weight symmetry or weight transport) are undesirable.

Interestingly, the distinct classes of locality also delineate distinct sets of experimentally testable predictions for plasticity. Critically, $\mathbf{S}p$-locality abstracts away details that are not important for testing predictions, and helps identify important features of learning algorithms. For example, the REINFORCE (Eq. 6) and Wake-Sleep (Eq. 8) algorithms are equivalent with respect to the '$p$' in $\mathbf{S}p$-locality, which tells us that the set of allowed local variables under $p$-locality itself (e.g. pre- and post-synaptic information) are not the best targets for experimental testing between these algorithms, and instead the focus must be on the variables included in '$\mathbf{S}$' (a reward signal $R$ versus a clamping variable $\gamma$ that switches the network between different modes of synaptic plasticity and neural activity). Thus, by identifying whether plasticity is modulated by reward, neuron-specific error teaching signals, transitions in network dynamics, or none of the above, one may narrow the space of possible candidate plasticity models down to $Rp$-, $\hat{\mathbf{e}}_i^{(l)}$-, $\gamma p_{md}$-, or $p_{(m)}$-locality, respectively. These distinct phenomena are *necessary* features of the learning algorithms in question for any neural network architecture used and are consequently good targets for experimental testing. In contrast, other features of parameter updates are more about specific architectural choices, e.g. switching from a rate-based network [16] to spiking [15], from feedforward [22] to recurrent [32], or single- to multi-compartment [38]. Thus, beyond clarifying and categorizing the locality properties of normative plasticity models, our hope is that our framework will also support experimental efforts to differentiate between such models, helping the field to focus in on the most important predictions to test experimentally.

# 5 Acknowledgments

BR was supported by NSERC (Discovery Grant: RGPIN-2020-05105; Discovery Accelerator Supplement: RGPAS-2020-00031; Arthur B. McDonald Fellowship: 566355-2022), CIFAR (Canada AI Chair; Learning in Machine and Brains Fellowship), the Canada Research Chair in Neural Computations and Interfacing. GL was supported by NSERC (Discovery Grant: RGPIN-2018-04821), CIFAR (Canada AI Chair), and a Canada Research Chair in Neural Computations and Interfacing. CB and GL acknowledge support from a Simon's Collaboration on the Global Brain Pilot Award. EW was supported by an NSERC CGS-D scholarship. CS was supported by the National Science Foundation under NSF Award No. 1922658.

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

# A  Defining Markov locality and relating it to $p$-locality

To gain intuition for how $p$-locality functions, we will introduce another notion of locality, called *Markov locality*, which will use the language of Markov blankets. We will prove that under relatively relaxed conditions $p$-locality and Markov locality are equivalent. This will allow us to relate the notion of locality to various graph structures commonly used to represent probability distributions, and will be a key step in proving Properties 2.1 and 2.2.

We start by defining the Markov boundary, $\mathcal{M}(X, S)$, of a random variable $X$ contained in a set of random variables $S$, as a minimal set such that $p(X|S) = p(X|\mathcal{M}(X, S))$. The Markov boundary defines a minimal set of variables such that, conditioned on these variables, conditioning on *no additional* random variables in $S$ changes the probability of $X$ [39]. Similarly, we define the Markov blanket, $M(X, S)$ for $X$ in $S$ as any set of variables such that conditioning on $M(X, S)$, makes $X$ conditionally independent from all other variables [39]. In this way, the Markov boundary is a Markov blanket but not all blankets are boundaries.

**Definition A.1.** *Markov locality: Given probability distribution $p(\mathbf{Z})$ and function $f : \mathbb{R}^{N_\mathbf{x} + N_\Theta} \to \mathbb{R}^{N_\Theta}$, the update function $f(\mathbf{Z})$ is Markov-local with respect to the distribution $p$ over $\mathbf{Z}$ if and only if $\forall\, k$:*

$$\exists\, \mathbf{Z} \in \Omega \text{ s.t. } \frac{\partial f_k(\mathbf{Z})}{\partial \mathbf{Z}_i} \neq 0 \Rightarrow \mathbf{Z}_i \in \mathcal{M}(\Theta_k, \mathbf{Z}). \tag{A.1}$$

Markov locality requires that the set of variables used in the parameter update $f_k(\mathbf{Z})$ is a subset of the Markov boundary of the parameter itself. A Markov boundary can be thought of as the set of variables that 'locally' communicate with the parameter $\Theta_k$, thus providing a natural measure of locality.

Importantly, for Markov-locality to be of use, we would like the Markov boundaries of random variables in the model of interest to be unique. Without this requirement there will be ambiguity, for a given $p$, in terms of which updates are considered local and which are not. To guarantee this, we ask that the conditional independence relationships implied by $p$ satisfy four properties, commonly referred to as graphoid properties [39, 40]. A sufficient condition for these to hold is that the distribution have a strictly positive density (see Appendix for more details B). With this, and some mild regularity assumptions, we can prove the following equivalence between Markov locality and $p$-locality:

**Theorem A.1.** *Assume all quantities are as in A.1, that the conditional independence relationships implied by $p(\mathbf{Z})$ satisfy the four graphoid properties given in Section B, and that mild regularity assumptions are satisfied by the joint distribution (see Section C.1). Then Equation 2 holds if and only if Equation A.1 also holds.*

*Proof.* This proof relies on Lemma A.1, proved below.

We wish to prove Eq. 2 $\iff$ Eq. A.1. It suffices to show the following:

$$\mathbb{E}_p\left[\left(\frac{\partial \log p(\mathbf{Z}_i | \mathbf{Z}_{\neq i})}{\partial \Theta_k}\right)^2\right] \neq 0 \iff \mathbf{Z}_i \in \mathcal{M}(\Theta_k, \mathbf{Z}) \tag{A.2}$$

Using the contrapositive for the left and right implications separately shows that Equation A.2 is equivalent to

$$\mathbb{E}_p\left[\left(\frac{\partial \log p(\mathbf{Z}_i | \mathbf{Z}_{\neq i})}{\partial \Theta_k}\right)^2\right] = 0 \iff \mathbf{Z}_i \notin \mathcal{M}(\Theta_k, \mathbf{Z}), \tag{A.3}$$

which means that it suffices to prove Equation A.3 for the proof. Observe that

$$E_p\left[\left(\frac{\partial \log p(\mathbf{Z}_i | \mathbf{Z}_{\neq i})}{\partial \Theta_k}\right)^2\right] = 0 \iff \frac{\partial \log p(\mathbf{Z}_i | \mathbf{Z}_{\neq i})}{\partial \mathbf{Z}_k} = 0 \quad \forall\, \mathbf{Z} \in \Omega, \tag{A.4}$$

which follows from the regularity assumptions. From here, the proof follows by Lemma A.1.

$\square$

**Lemma A.1.** *Let* $\mathbf{X}$, $\boldsymbol{\Theta}$, $\mathbf{Z}$, $k$ *and* $i$, *and* $p$ *be defined as in Theorem A.1. Then*

$$\frac{\partial \log p(\mathbf{Z}_i | \mathbf{Z}_{\neq i})}{\partial \mathbf{Z}_k} = 0 \quad \forall \, \mathbf{Z} \in \Omega \iff \mathbf{Z}_i \notin \mathcal{M}(\mathbf{Z}_k, \mathbf{Z}) \tag{A.5}$$

*Proof.* First, observe that

$$\frac{\partial \log p(\mathbf{Z}_i | \mathbf{Z}_{\neq i})}{\partial \mathbf{Z}_k} = 0 \quad \forall \quad \mathbf{Z} \in \Omega$$

$$\iff \frac{\partial p(\mathbf{Z}_i | \mathbf{Z}_{\neq i})}{\partial \mathbf{Z}_k} = 0 \quad \forall \quad \mathbf{Z} \in \Omega \tag{A.6}$$

by the chain rule. By applying the fundamental theorem of calculus to this derivative, which we can do by the assumption of differentiability on $\mathbb{R}$, we find that $p(\mathbf{Z}_i | \mathbf{Z}_{\neq i})$ is constant w.r.t. $\mathbf{Z}_k$ on $\Omega$ so that

$$p(\mathbf{Z}_i | \mathbf{Z}_{\neq i}) = p(\mathbf{Z}_i | \mathbf{Z}_{\neq i}) \int_{\mathbb{R}} p(\mathbf{Z}_k | \mathbf{Z}_{\neq \{i,k\}}) \mathrm{d}\mathbf{Z}_k \tag{A.7}$$

$$= \int_{\mathbb{R}} p(\mathbf{Z}_i, \mathbf{Z}_k | \mathbf{Z}_{\neq \{i,k\}}) \mathrm{d}\mathbf{Z}_k = p(\mathbf{Z}_i | \mathbf{Z}_{\neq \{i,k\}}) \tag{A.8}$$

where we have also used that a probability distribution integrates to 1, and that $p(\mathbf{Z}_k | \mathbf{Z}_{\neq \{i,k\}})$ will be equal to zero outside $\Omega$. Because $\mathbf{Z} \in \Omega$ is arbitrary, from the above $\mathbf{Z}_i$ is independent of $\mathbf{Z}_k$ given the other random variables in $\mathbf{Z}$. Using Lemma B.1 (which we can do by assumption of the graphoid properties), we see that if $\frac{\partial \log p(\mathbf{Z}_i | \mathbf{Z}_{\neq i})}{\partial \mathbf{Z}_k} = 0 \quad \forall \quad \mathbf{Z} \in \Omega$, by Eq. A.6 - A.8, $\mathbf{Z}_i \notin \mathcal{M}(\mathbf{Z}_k, \mathbf{Z})$. Conversely, if we start with the assumption that $\mathbf{Z}_i \notin \mathcal{M}(\mathbf{Z}_k, \mathbf{Z})$, we immediately get $\mathbf{Z}_k \notin \mathcal{M}(\mathbf{Z}_i, \mathbf{Z})$, by Lemma B.1, and see that $p(\mathbf{Z}_i | \mathbf{Z}_{\neq i})$ must not be a function of $\mathbf{Z}_k$ for all $\mathbf{Z}$; thus, the derivative w.r.t. $\mathbf{Z}_k$ is equal to zero for $\mathbf{Z} \in \Omega$. Applying Equation A.6 completes the proof.

$\square$

## B Notes on Probabilistic Graphical Models

In this section we compile several properties, definitions, and results on Markov boundaries and Probabilistic Graphical Models (PGMs) that underlie Theorem A.1, and Properties 2.1 and 2.2. We begin by setting up notation. Let us assume that we have a joint probability distribution, $P$, over a set of random variables $S$, and that $W, X, Y, Z \subset S$, and $U, V, T \in S$. We use $X \perp\!\!\!\perp Y | Z$ to mean that the set of random variables $X$ is independent of the set $Y$ given set $Z$, and assume that the reader is familiar with the notion of a directed graph, an undirected graph, and graph separation. If a set $X$ contains only a single random variable $U$ then we abuse notation and write $U$ in place of $X$.

The following four properties–known as the *graphoid properties* (or axioms–see e.g. [39, 40])–are useful for getting well-behaved Markov boundaries and in assigning graphical representation to probability distributions:

**Definition B.1.** *Pseudo-graphoid properties:*

- *Symmetry:* $X \perp\!\!\!\perp Y | Z \implies Y \perp\!\!\!\perp X | Z$

- *Decomposition:* $X \perp\!\!\!\perp Y, W | Z \implies X \perp\!\!\!\perp Y | Z \, \& \, X \perp\!\!\!\perp W | Z$

- *Weak union:* $X \perp\!\!\!\perp Y, W | Z \implies X \perp\!\!\!\perp Y | W, Z$

- *Intersection:* $X \perp\!\!\!\perp Y | Z, W \, \& \, X \perp\!\!\!\perp W | Z, Y \implies X \perp\!\!\!\perp Y, W | Z$

Importantly, it is known that these properties are satisfied when we have a density $p$ that is strictly positive w.r.t. to its base product measure [40]. Here, measure is used in a measure theoretic sense; e.g. we have assumed throughout the paper that the base measure is simply a product of a multi-dimensional Lebesgue measure over $\mathbb{R}^{k_1}$, for some $k_1$, and a counting measure over $\mathbb{N}^{k_2}$ or some subset of $\mathbb{N}^{k_2}$, for some $k_2$. Roughly speaking, this positivity property means that there are no purely deterministic relationships between variables.

The key result that we use to guarantee that the Markov boundaries we discuss in the paper are well-defined is given in [39]. We state a paraphrased and shortened version below for completeness:

**Theorem B.1.** *Theorem 4, Chapter 3 in [39]: every $P$ with conditional independence relations satisfying the four pseudo-graphoid properties has a unique Markov boundary for each $X$.*

We now add two more simple results on Markov boundaries, used in the proof of Theorem A.1:

**Lemma B.1.** *If $P$ has conditional independence relations satisfying the four pseudo-graphoid properties we have:*

- *for every $U, V \in S$, $U \in \mathcal{M}(V, S) \iff V \in \mathcal{M}(U, S)$*

- *for every $U \in S$, $\mathcal{M}(U, S)$ is contained in every Markov blanket of $U$.*

These follow simply from the graphoid properties so we omit the proof.

Lastly, we make specific what we mean when we say that a graph is an undirected or directed graphical model for a distribution.

**Definition B.2.** *Let $\mathcal{G}$ be an undirected graph where each node corresponds to a random variable in $S$. We say that $\mathcal{G}$ is an **Undirected Graph (UG) for** $\mathbf{P}$ if whenever $X$ and $Y$ are separated by $Z$ in $\mathcal{G}$, $X \perp\!\!\!\perp Y | Z$ is true under $P$. Note that this corresponds to the notion of I-map in [39].*

**Definition B.3.** *Let $\mathcal{G}_{\mathrm{d}}$ be a directed graph with each vertex corresponding to a random variable in $S$. We say that $\mathcal{G}_{\mathrm{d}}$ **is a Directed Graph for** $\mathbf{P}$ if the variable under $P$ corresponding to any node in the graph is conditionally independent of all variables corresponding to nodes that are non-descendants given the variables corresponding to parents. This is equivalent to $\mathcal{G}_{\mathrm{d}}$ satisfying the Markov condition described in Definition 1.9 of [41].*

## C  Proofs for $p$-locality properties

For the first two properties we assume the requirements of Theorem A.1 are satisfied. For all properties *except* 2.3, 2.4, and 2.8 we assume $p$ satisfies mild regularity constraints (see Section C.1).

**Property 2.1** Assume $\mathcal{G}_{\mathrm{d}}$ is a Directed Acyclic Graph (DAG) for $p$. If $\frac{\partial f_k(\mathbf{Z})}{\partial \mathbf{Z}_i} \neq 0$ for $\mathbf{Z}_i$ that is not a parent, co-parent, or child of $\mathbf{\Theta}_k$ in $\mathcal{G}_{\mathrm{d}}$, then $f$ is not $p$-local.

*Proof.* By Theorem A.1 we get that $\frac{\partial f_k(\mathbf{Z})}{\partial \mathbf{Z}_i}$ can only be non-zero on the unique Markov boundary of $\mathbf{Z}_i$ if it is $p$-local. By the definition of a DAG, the parents, co-parents, and children of $\mathbf{Z}_i$ form a Markov blanket for it (see e.g. [41] Th. 2.13). By Lemma B.1 the boundary is included in all Markov blankets so $\frac{\partial f_k(\mathbf{Z})}{\partial \mathbf{Z}_i}$ can only be non-zero on some subset of the parents, co-parents, and children of $\mathbf{Z}_i$. $\square$

**Property 2.2** Assume that $\mathcal{G}$ defines an Undirected Graph (UG) for $p$. If $\frac{\partial f_k(\mathbf{Z})}{\partial \mathbf{Z}_i} \neq 0$ for $\mathbf{Z}_i$ that is not a neighbour of $\mathbf{\Theta}_k$ in $\mathcal{G}$, then $f$ is not $p$-local.

*Proof.* As above, by Theorem A.1 we get that $\frac{\partial f_k(\mathbf{Z})}{\partial \mathbf{Z}_i}$ can only be non-zero on the unique Markov boundary of $\mathbf{Z}_i$ if it is $p$-local. A UG for a distribution is an I-map for it, and conditioning on the neighbours in an I-map renders a node independent from the other nodes in the graph by definition–thus the neighbours form a Markov blanket. By Lemma B.1 the Markov boundary is included in every blanket so $\frac{\partial f_k(\mathbf{Z})}{\partial \mathbf{Z}_i}$ can only be non-zero on some subset of the neighbours in the UG. $\square$

**Property 2.3** *For any function* $b(\mathbf{Z}) : \mathbb{R}^{N_\mathbf{x}+N_\Theta} \rightarrow \mathbb{R}^{N_\Theta}$ *defined such that* $b_k(\mathbf{Z}) = h_k(f_k(\mathbf{Z}), g_k(\mathbf{Z}))$, *where* $f$ *and* $g$ *are p-local and* $h_k$ *is differentiable,* $b(\mathbf{Z})$ *is also p-local.*

*Proof.* Suppose that $\frac{\partial b_k(\mathbf{Z})}{\partial \mathbf{X}_i} \neq 0$. We need to show that $\mathbb{E}_p\left[\left(\frac{\partial \log p(\mathbf{Z}_i|\mathbf{Z}_{\neq i})}{\partial \Theta_k}\right)^2\right] \neq 0$. Knowing that $\frac{\partial b_k(\mathbf{Z})}{\partial \mathbf{X}_i} \neq 0$, we have:

$$\frac{\partial h_k(\mathbf{Z})}{\partial f_k}\frac{\partial f_k(\mathbf{Z})}{\partial \mathbf{Z}_i} + \frac{\partial h_k(\mathbf{Z})}{\partial g_k}\frac{\partial g_k(\mathbf{Z})}{\partial \mathbf{X}_i} \neq 0. \tag{C.1}$$

This implies that either $\frac{\partial f_k(\mathbf{Z})}{\partial \mathbf{Z}_i} \neq 0$ or $\frac{\partial g_k(\mathbf{Z})}{\partial \mathbf{Z}_i} \neq 0$ (or both). No matter which is true, by virtue of the $p$-locality of $f$ and $g$, we have the consequence:

$$\mathbb{E}_p\left[\left(\frac{\partial \log p(\mathbf{Z}_i|\mathbf{Z}_{\neq i})}{\partial \Theta_k}\right)^2\right] \neq 0, \tag{C.2}$$

which concludes our proof. This demonstrates that $p$-local functions can be more or less arbitrarily combined without the combination losing the $p$-local property. $\square$

**Property 2.4** *For any function* $f(\cdot) : \mathbb{R}^{N_\mathbf{x}+N_\Theta} \rightarrow \mathbb{R}^{N_\Theta}$, *there exists a probability distribution* $p(\mathbf{Z})$ *such that the random variable* $f(\mathbf{Z})$ *with* $\mathbf{Z} \sim p(\mathbf{Z})$ *is p-local.*

*Proof.* We can prove this property by construction. Take $p(\{\mathbf{X}_i : \frac{\partial f(\mathbf{X})}{\partial \mathbf{X}_i} \neq 0\}|\Theta) = \mathcal{N}(\Theta^T\Theta, \mathbf{I})$, i.e. the distribution of every variable contained within $f$ has mean parameter dependence on all $\Theta$ variables. The probability distributions for all other variables $\mathbf{Z}$ are otherwise unconstrained. Then for all $i$ such that $\frac{\partial f(\mathbf{X})}{\partial \mathbf{X}_i} \neq 0$, we have:

$$\mathbb{E}_p\left[\left(\frac{\partial \log p(\mathbf{X}_i|\mathbf{X}_{\neq i}, \Theta)}{\partial \Theta_k}\right)^2\right] = \mathbb{E}_p\left[\left(\frac{\partial \log p(\mathbf{X}_i|\Theta)}{\partial \Theta_k}\right)^2\right] \tag{C.3}$$

$$= \mathbb{E}_p\left[\left(-\frac{\partial}{\partial \Theta_k}\frac{(\mathbf{X}_i - \Theta^T\Theta)^2}{2}\right)^2\right] \tag{C.4}$$

$$= \mathbb{E}_p\left[2\left((\mathbf{X}_i - \Theta^T\Theta)\Theta_k\right)^2\right] \tag{C.5}$$

$$= 4\mathbb{E}_{p(\Theta)}\left[\Theta_k^2\mathbb{E}_{p(\mathbf{X}_i|\Theta)}\left[(\mathbf{X}_i - \Theta^T\Theta)^2\right]\right] \tag{C.6}$$

$$= 4\mathbb{E}_{p(\Theta)}\left[\Theta_k^2\right] \neq 0. \tag{C.7}$$

$\square$

**Property 2.5** *The derivative of the log joint distribution* $\frac{\partial \log p(\mathbf{X},\Theta)}{\partial \Theta}$ *is p-local.*

Here, it's more useful to work with the equivalent (contrapositive) requirement for $p$-locality, i.e., we need to show $\forall k, i$:

$$\mathbb{E}_p\left[\left(\frac{\partial \log p(\mathbf{Z}_i|\mathbf{Z}_{\neq i})}{\partial \Theta_k}\right)^2\right] = 0 \Rightarrow \frac{\partial^2 \log p(\mathbf{X}, \Theta)}{\partial \mathbf{Z}_i \partial \Theta_k} = 0. \tag{C.8}$$

*Proof.* First, we see that:

$$\mathbb{E}_p\left[\left(\frac{\partial \log p(\mathbf{Z}_i|\mathbf{Z}_{\neq i})}{\partial \Theta_k}\right)^2\right] = 0 \tag{C.9}$$

$$\Rightarrow \frac{\partial \log p(\mathbf{Z}_i|\mathbf{Z}_{\neq i})}{\partial \Theta_k} = 0 \tag{C.10}$$

$$\Rightarrow \frac{\partial^2 \log p(\mathbf{Z}_i|\mathbf{Z}_{\neq i})}{\partial \mathbf{Z}_i \partial \Theta_k} = 0, \tag{C.11}$$

where the first implication follows from the fact that the Fisher Information integral is effectively a weighted sum of elements, each of which is $\geq 0$. If the function on the right were nonzero for some $\mathbf{Z}$, then the Fisher Information would also be nonzero. Assuming that $\log p$ has differentiable partial derivatives, we can interchange the order of differentiation, giving:

$$\Rightarrow \frac{\partial^2 \log p(\mathbf{Z}_i|\mathbf{Z}_{\neq i})}{\partial \mathbf{\Theta}_k \partial \mathbf{Z}_i} = 0 \tag{C.12}$$

$$\Rightarrow \frac{\partial}{\partial \mathbf{\Theta}_k} \left[ \frac{\partial}{\partial \mathbf{Z}_i} \left[ \log p(\mathbf{Z}_i|\mathbf{Z}_{\neq i}) + \log p(\mathbf{Z}_{\neq i}) \right] \right] = 0 \tag{C.13}$$

$$\Rightarrow \frac{\partial}{\partial \mathbf{\Theta}_k} \left[ \frac{\partial}{\partial \mathbf{Z}_i} \left[ \log p(\mathbf{Z}) \right] \right] = 0 \tag{C.14}$$

$$\Rightarrow \frac{\partial^2 \log p(\mathbf{Z})}{\partial \mathbf{Z}_i \partial \mathbf{\Theta}_k} = 0, \tag{C.15}$$

which concludes the proof. □

**Property 2.6** *For a probability distribution given by* $p(\mathbf{Z}) = \frac{1}{\mathcal{Z}} \exp\left(-E(\mathbf{Z})\right)$, *the expression* $\frac{\partial}{\partial \mathbf{\Theta}} E(\mathbf{Z})$ *is p-local.*

*Proof.* The proof is almost identical to the proof for Property 2.5. From Property 2.5, we have that:

$$\mathbb{E}_p \left[ \left( \frac{\partial \log p(\mathbf{Z}_i|\mathbf{Z}_{\neq i})}{\partial \mathbf{\Theta}_k} \right)^2 \right] = 0 \tag{C.16}$$

$$\Rightarrow \frac{\partial^2 \log p(\mathbf{Z})}{\partial \mathbf{Z}_i \partial \mathbf{\Theta}_k} = 0. \tag{C.17}$$

Using our definition of $p$, we have:

$$\Rightarrow \frac{-\partial^2 \left( E(\mathbf{Z}) + \log \mathcal{Z} \right)}{\partial \mathbf{Z}_i \partial \mathbf{\Theta}_k} = 0 \tag{C.18}$$

$$\Rightarrow \frac{\partial^2 E(\mathbf{Z})}{\partial \mathbf{Z}_i \partial \mathbf{\Theta}_k} = 0, \tag{C.19}$$

which concludes the proof. □

**Property 2.7** *If the marginal parameter distribution factorizes as* $p(\mathbf{\Theta}) = \prod_k p(\mathbf{\Theta}_k)$, *i.e. the parameters are independent from one another, then the score function* $\frac{\partial \log p(\mathbf{X}|\mathbf{\Theta})}{\partial \mathbf{\Theta}}$ *is p-local.*

*Proof.* Again, we make heavy use of Property 2.5, which states:

$$\mathbb{E}_p \left[ \left( \frac{\partial \log p(\mathbf{Z}_i|\mathbf{Z}_{\neq i}, \mathbf{\Theta})}{\partial \mathbf{\Theta}_k} \right)^2 \right] = 0 \Rightarrow \frac{\partial^2 \log p(\mathbf{Z})}{\partial \mathbf{Z}_i \partial \mathbf{\Theta}_k} = 0. \tag{C.20}$$

It is important to note that the left-hand equation only holds true if $\mathbf{Z}_i \neq \mathbf{\Theta}_k$: under $p$-locality, an update equation for parameter $\mathbf{\Theta}_k$ can always include its own value. So for the remainder of the proof we will assume that $\mathbf{Z}_i \neq \mathbf{\Theta}_k$. Now, $\log(p(\mathbf{X}|\mathbf{\Theta})) = \log p(\mathbf{Z}) - \log p(\mathbf{\Theta})$. We also have:

$$\frac{\partial^2 \log p(\mathbf{\Theta})}{\partial \mathbf{Z}_i \partial \mathbf{\Theta}_k} = \frac{\partial^2 \sum_k \log p(\mathbf{\Theta}_k)}{\partial \mathbf{Z}_i \partial \mathbf{\Theta}_k} = 0, \tag{C.21}$$

where for the last equality we have used the assumption that $\mathbf{Z}_i \neq \mathbf{\Theta}_k$. These two equations collectively imply:

$$\Rightarrow \frac{\partial^2 \log p(\mathbf{X}|\mathbf{\Theta})}{\partial \mathbf{Z}_i \partial \mathbf{\Theta}_k} = \frac{\partial^2 \log p(\mathbf{Z}) - \log p(\mathbf{\Theta})}{\partial \mathbf{Z}_i \partial \mathbf{\Theta}_k} = 0, \tag{C.22}$$

which concludes the proof. □

**Property 2.8** *For a mixture distribution $p_{12}(\mathbf{Z}, \gamma) = p_1(\mathbf{Z})^\gamma p_2(\mathbf{Z})^{1-\gamma} p(\gamma)$ for some binary variable $\gamma \in \{0, 1\}$ with nonzero probabilities, if $f(\mathbf{Z})$ is $p_1$-local (or equivalently $p_2$-local), then $f(\mathbf{Z})$ is $p_{12}$-local.*

*Proof.* We again work with the contrapositive definition of $p$-locality, observing that:

$$\mathbb{E}_{p_{12}(\mathbf{Z}, \gamma)} \left[ \left( \frac{\partial \log p_{12}(\mathbf{Z}_i | \mathbf{Z}_{\neq i})}{\partial \mathbf{\Theta}_k} \right)^2 \right] = 0 \tag{C.23}$$

$$\Rightarrow \sum_{k \in \{0,1\}} p(\gamma = k) \mathbb{E}_{p_k(\mathbf{Z})} \left[ \left( \frac{\partial \log p_k(\mathbf{Z}_i | \mathbf{Z}_{\neq i})}{\partial \mathbf{\Theta}_k} \right)^2 \right] = 0 \tag{C.24}$$

$$\Rightarrow \mathbb{E}_{p_1(\mathbf{X}; \mathbf{\Theta})} \left[ \left( \frac{\partial \log p_1(\mathbf{Z}_i | \mathbf{Z}_{\neq i})}{\partial \mathbf{\Theta}_k} \right)^2 \right] = 0 \tag{C.25}$$

$$\Rightarrow \frac{f_k(\mathbf{X})}{\partial \mathbf{Z}_i} = 0, \tag{C.26}$$

where the third implication follows from the fact that if the sum of two nonnegative quantities is zero, then *both* quantities are zero, and the final implication holds from the $p_1$-locality of $f(\mathbf{Z})$. $\square$

## C.1 Regularity of Joint distribution

For several of these properties we enforce mild regularity constraints on the density. This is because we want the integral of the squared score being equal to zero to imply that the score itself is equal to zero. A sufficient condition for this is that the joint density function and partial derivatives w.r.t. $\mathbf{\Theta}_k \ \forall k$ are, for every fixed value of $\mathbf{Z}$'s discrete elements, continuous functions of $\mathbf{Z}$'s continuous elements.

# D  Locality for a linear continuous Boltzmann machine

Consider the following example of a simplified linear recurrent neuron model with synaptic weight matrices $\mathbf{W}$. The joint distribution is given by:

$$p(\mathbf{r}, \mathbf{W}) = \frac{1}{Z} e^{-E(\mathbf{r}, \mathbf{W})} \tag{D.1}$$

$$E(\mathbf{r}, \mathbf{W}) = \left( \frac{1}{2\tau} \|\mathbf{r}\|_2^2 - \frac{1}{2} \mathbf{r}^T \mathbf{W} \mathbf{r} + \frac{1}{2} \|\mathbf{W}\|_2^2 \right) / \sigma^2 \tag{D.2}$$

$$= \left( \frac{1}{2\tau} \sum_i \mathbf{r}_i^2 - \frac{1}{2} \sum_{ij} \mathbf{W}_{ij} \mathbf{r}_i \mathbf{r}_j + \frac{1}{2} \sum_{ij} \mathbf{W}_{ij}^2 \right) / \sigma^2, \tag{D.3}$$

where $\mathbf{W}$ is assumed to be a symmetric matrix. To see why this probability distribution is relevant for neuroscience, we first note that $E$ is a linear, continuous analog of the Hopfield energy function, which is also used for discrete-valued Boltzmann machines. There are two critical differences between this probability distribution and the linear feedforward network explored in Section 2.4: first, this distribution corresponds to the *undirected graphical model* shown in Figure 1c, as opposed to the DAG shown in Figure 1a; second, the marginal distribution is not a free parameter that we can choose with convenient factorization properties if we want our joint distribution to give us a version of $p$-locality that corresponds with our concept of biological locality. For undirected graphical models, one typically is required to define the joint distribution first, and compute conditional distributions explicitly through Bayes theorem or approximate through some form of MCMC sampling. In our case, we can see that $p(\mathbf{r} | \mathbf{W})$ corresponds to the steady-state distribution of a stochastic differential equation using $E$ to perform Langevin sampling:

$$dr_i = - \left[ \nabla_{\mathbf{r}_i(t)} E(\mathbf{r}_i(t), \mathbf{W}) \sigma^2 \right] dt + \sigma d\mathbf{B}_i(t) \tag{D.4}$$

$$= \left[ -\frac{1}{\tau} \mathbf{r}_i(t) + \sum_j \mathbf{W}_{ij} \mathbf{r}_j(t) \right] dt + \sigma d\mathbf{B}_i(t), \tag{D.5}$$

where here $\mathbf{B}_i(t)$ corresponds to uncorrelated Brownian noise injected into the system. These stochastic sampling dynamics correspond to a noisy linear recurrent network. Therefore, $p(\mathbf{r}|\mathbf{W})$ corresponds to the steady-state stimulus response distribution of a linear recurrent network.

Let's ask: for which neural indices $k$ can we have $\frac{\partial}{\partial \mathbf{r}_k} f_{\mathbf{W}_{ij}}(\mathbf{r}) \neq 0$ so that the function $f$ is still $p$-local? For $f$ to remain $p$-local, we would need $\mathbb{E}_p \left[ \left( \frac{\partial \log p(\mathbf{r}_k | \mathbf{r}_{\neq k}, \mathbf{W})}{\partial \mathbf{W}_{ij}} \right)^2 \right] \neq 0$. For the definition of $p$-locality to conform to our intuitions about biological locality, we would expect the only allowable variables to be the pre- and postsynaptic neurons $\mathbf{r}_i$ and $\mathbf{r}_j$—we will show that including any other variable will violate $p$-locality. To see why, suppose $k \neq i, j$. Note that we can decompose $E$ as:

$$E(\mathbf{r}, \mathbf{W}) = E_k + E_{\neq k} \tag{D.6}$$

$$E_k = \left( \frac{1}{2\tau} \mathbf{r}_k^2 - \frac{1}{2} \sum_j \mathbf{W}_{kj} \mathbf{r}_k \mathbf{r}_j - \frac{1}{2} \sum_j \mathbf{W}_{jk} \mathbf{r}_j \mathbf{r}_k \right) / \sigma^2 \tag{D.7}$$

$$E_{\neq k} = \left( \frac{1}{2\tau} \sum_{i \neq k} \mathbf{r}_i^2 - \frac{1}{2} \sum_{ij \neq k} \mathbf{W}_{ij} \mathbf{r}_i \mathbf{r}_j + \frac{1}{2} \sum_{ij} \mathbf{W}_{ij}^2 \right) / \sigma^2. \tag{D.8}$$

Under this decomposition, $E_k$ has no dependency on $\mathbf{W}_{ij}$, and $E_{\neq k}$ has no dependency on $\mathbf{r}_k$. Now we're in a position to demonstrate that for any choice of $k$ such that $k \neq i, j$, $f_{\mathbf{W}_{ij}}(\mathbf{r})$ *cannot* be $p$-local.

$$p(\mathbf{r}_k | \mathbf{r}_{\neq k}, \mathbf{W}) = \frac{p(\mathbf{r}|\mathbf{W})}{p(\mathbf{r}_{\neq k}, \mathbf{W})} \tag{D.9}$$

$$= \frac{p(\mathbf{r}|\mathbf{W})}{\int p(\mathbf{r}, \mathbf{W}) d\mathbf{r}_k} \tag{D.10}$$

$$= \frac{e^{-E}}{\int e^{-E} d\mathbf{r}_k} \tag{D.11}$$

$$= \frac{e^{-(E_k + E_{\neq k})}}{e^{-E_{\neq k}} \int e^{-(E_k)} d\mathbf{r}_k} \tag{D.12}$$

$$= \frac{e^{-(E_k)}}{\int e^{-(E_k)} d\mathbf{r}_k} \tag{D.13}$$

$$\Rightarrow \frac{\partial}{\partial \mathbf{W}_{ij}} \log p(\mathbf{r}_k | \mathbf{r}_{\neq k}, \mathbf{W}) = 0 \tag{D.14}$$

$$\Rightarrow \mathbb{E}_p \left[ \left( \frac{\partial \log p(\mathbf{r}_k | \mathbf{r}_{\neq k}, \mathbf{W})}{\partial \mathbf{W}_{ij}} \right)^2 \right] = 0. \tag{D.15}$$

Because the conditional distribution has no dependency on $\mathbf{W}_{ij}$, then the Fisher Information is also 0, which concludes the demonstration: $f_{\mathbf{W}_{ij}}$ is not $p$-local if it is a function of $r_k$ for $k \neq i, j$. Of course, this decomposition of $E = E_k + E_{\neq k}$ would not be possible if $k = i$ or $j$. To summarize, for our simple example, any parameter update for $\mathbf{W}_{ij}$ that depends on the activity of any neuron $\mathbf{r}_k$ that is not the pre- or postsynaptic neuron ($k \neq i, j$) cannot be $p$-local. Alternatively, since this is an undirected graphical model, we can also inspect its corresponding graph (summarized in Figure 1c.). To verify that the graph in Figure 1c. corresponds to our network, observe that our probability

distribution factorizes according the cliques of the graph [42] as follows:

$$p(\mathbf{r}, \mathbf{W}) = \frac{1}{\mathcal{Z}} \prod_i \phi(\mathbf{r}_i) \prod_{ij} \phi(\mathbf{W}_{ij}) \prod_{ij} \phi(\mathbf{r}_i, \mathbf{r}_j, \mathbf{W}_{ij}) \tag{D.16}$$

$$\phi(\mathbf{r}_i) = \exp\left(\frac{1}{2\tau\sigma^2}\mathbf{r}_i^2\right) \tag{D.17}$$

$$\phi(\mathbf{W}_{ij} = \exp\left(\frac{1}{2\sigma^2}\mathbf{W}_{ij}^2\right) \tag{D.18}$$

$$\phi(\mathbf{r}_i, \mathbf{r}_j, \mathbf{W}_{ij}) = \exp\left(-\frac{1}{2\sigma^2}\mathbf{W}_{ij}\mathbf{r}_i\mathbf{r}_j\right). \tag{D.19}$$

Looking at the graph, we can verify by inspection that the only neighbors of $\mathbf{W}_{ij}$ are $\mathbf{r}_i$ and $\mathbf{r}_j$, which confirms our detailed analysis by Property 2.2.

# E  Proofs of $p$-locality properties of normative plasticity algorithms

## E.1  REINFORCE

**Theorem E.1.** *If $p(\Theta) = \prod_k p(\Theta_k)$, the REINFORCE estimator given by $\mathcal{A}_R(p(R, \mathbf{X}|\Theta))$ is $Rp$-local.*

*Proof.* The REINFORCE derivation proceeds as follows: suppose that we have some probabilistic formulation of a neural network and incoming sensory stimuli $p(\mathbf{X}|\Theta)$ and some probabilistic reward function $p(R|\mathbf{X})$ dependent on the stimuli and neural responses. We want to maximize expected reward:

$$\mathbb{E}[R] = \int R p(R|\mathbf{X}) p(\mathbf{X}|\Theta) d\mathbf{X} dR. \tag{E.1}$$

If we want to modify our parameters $\Theta$ in order to improve performance, we take steps in an approximation of the direction of the gradient of the objective $\mathbb{E}[R]$.

$$\frac{\partial}{\partial \Theta}\mathbb{E}[R] = \frac{\partial}{\partial \Theta} \int R p(R|\mathbf{X}) p(\mathbf{X}|\Theta) d\mathbf{X} dR \tag{E.2}$$

$$= \int R p(R|\mathbf{X}) \frac{\partial}{\partial \Theta} p(\mathbf{X}|\Theta) d\mathbf{X} dR \tag{E.3}$$

$$= \int R p(R|\mathbf{X}) \frac{\partial}{\partial \Theta} e^{\log p(\mathbf{X}|\Theta)} d\mathbf{X} dR \tag{E.4}$$

$$= \int R p(R|\mathbf{X}) \left[\frac{\partial}{\partial \Theta} \log p(\mathbf{X}|\Theta)\right] p(\mathbf{X}|\Theta) d\mathbf{X} dR \tag{E.5}$$

$$\approx \frac{1}{K} \sum_{k=1}^{K} R^{(k)} \frac{\partial}{\partial \Theta} \log p(\mathbf{X}^{(k)}|\Theta), \tag{E.6}$$

where in this last step we have employed a Monte Carlo approximation of the expectation, where $R^{(k)}$ and $\mathbf{X}^{(k)} \sim p(R, \mathbf{X})$. This update function: $f(R, \mathbf{Z}) = R \times \frac{\partial}{\partial \Theta} \log p(\mathbf{X}|\Theta)$ is not $p$-local because we have $\partial f(R, \mathbf{Z})/\partial R \neq 0$, while $\frac{\partial}{\partial \Theta} p(R|\mathbf{X}) = 0$. However, as we know, this update is the product of a score function with a marginal parameter distribution that we have assumed factorizes, which we know to be $p$-local by Property 2.7, with a scalar reward $R$. In this case, one could postulate that reward information is projected broadly to many synapses in the neural network via a neuromodulatory pathway (Figure 2a). We see that $f(R, \mathbf{X}) = h(R, g(\mathbf{X}))$ if we take $h(a, b) = a \times b$ and $g(\mathbf{X}) = \frac{\partial}{\partial \Theta} \log p(\mathbf{X}|\Theta)$; we further see that $g$ is $p$-local, and hence $f(R, \mathbf{X})$ is by Definition 2.3 $Rp$-local.

$\square$

This might seem contrived, because *any* function is $\mathbf{S}p$-local for some sufficiently broad choice of $\mathbf{S}$. However, we have shown here that the REINFORCE algorithm is $Rp$-local for *any* choice of $p$ with a marginal parameter distribution that factorizes (an easy constraint to satisfy for directed graphical model architectures). This means that we can make any of a huge variety of neural network or probabilistic model choices and still have the REINFORCE algorithm obey the same notion of locality, without having to modify our definition post-hoc.

## E.2  Maximum Likelihood Estimation (MLE)

MLE is a highly popular machine learning method and the fundamental basis for several subsequent normative plasticity algorithms. This algorithm involves fitting a model, $p_m(\mathbf{X}|\boldsymbol{\Theta})$, to an empirical data distribution, $p_d(\mathbf{X})$.

**Theorem E.2.** *If $p(\boldsymbol{\Theta}) = \prod_k p(\boldsymbol{\Theta}_k)$, the MLE update given by $\mathcal{A}_{MLE}(p_m(\mathbf{X}|\boldsymbol{\Theta}), p_d(\mathbf{X}))$ is $p_m$-local.*

*Proof.* We proceed by first deriving the MLE update function. The objective function for maximum likelihood estimation is given by the KL divergence between an empirical data distribution, $p_d(\mathbf{X})$ and a probabilistic model of the data $p_m(\mathbf{X}|\boldsymbol{\Theta})$. We have:

$$\mathrm{KL}[p_d(\mathbf{X})||p_m(\mathbf{X}|\boldsymbol{\Theta})] = -\int \log\left(\frac{p_m(\mathbf{X}|\boldsymbol{\Theta})}{p_d(\mathbf{X})}\right)p_d(\mathbf{X})d\mathbf{X}. \tag{E.7}$$

We want to minimize this objective function, which we do by gradient descent:

$$\mathcal{A}_{MLE}(p(\mathbf{X}), p_m(\mathbf{X})) \propto \frac{\partial}{\partial\boldsymbol{\Theta}}\int \log\left(\frac{p_m(\mathbf{X}|\boldsymbol{\Theta})}{p_d(\mathbf{X})}\right)p_d(\mathbf{X})d\mathbf{X} \tag{E.8}$$

$$= \int \frac{\partial}{\partial\boldsymbol{\Theta}}\log\left(p_m(\mathbf{X}|\boldsymbol{\Theta})\right)p_d(\mathbf{X})d\mathbf{X} \tag{E.9}$$

$$\approx \frac{1}{K}\sum_{k=0}^{K}\frac{\partial}{\partial\boldsymbol{\Theta}}\log\left(p_m(\mathbf{X}_k|\boldsymbol{\Theta})\right), \tag{E.10}$$

where $\mathbf{X}_k \sim p_d(\mathbf{X})$, and in the last approximate equality we have used a Monte Carlo sampling integral approximation. This update exclusively contains the score function of $p_m$, so by Property 2.7, the update is $p_m$-local. $\qquad\square$

## E.3  Generalized EM (GEM)

MLE estimation runs into difficulties when attempting to fit latent variable models, (e.g. when $p_m(\mathbf{X}_o) = \int p_m(\mathbf{X}_o, \mathbf{X}_h)d\mathbf{X}_h$), where $\mathbf{X}_h$ are latent variables within the model distribution that 'explain' observed data $\mathbf{X}_o$. Latent variable models are extraordinarily powerful, and appear in computational neuroscience in a variety of forms, including but not limited to factor analysis, hidden Markov models, and Kalman Filters [43]; for these models, we will take $\mathbf{X} = [\mathbf{X}_o, \mathbf{X}_h]$. Instead of performing explicit MLE, when fitting latent variable models one usually resorts to some variant of the Expectation-Maximization (EM) algorithm [44]. Here, we show that a particular variant of the EM algorithm, called Generalized EM (GEM) [45], is $p_m$-local in the same way as MLE.

GEM gains computational benefits by substituting (by any of a variety of methods) an approximate posterior distribution $p_d(\mathbf{X}_h|\mathbf{X}_o)$ for the true, but typically intractable, model posterior $p_m(\mathbf{X}_h|\mathbf{X}_o)$ via minimizing a variational free energy [45]. However, GEM is not just a convenient model-fitting algorithm: in subsequent sections, we will show that the $p_m$-locality of GEM explains why several popular normative plasticity algorithms produce biologically plausible updates.

**Theorem E.3.** *If $p(\boldsymbol{\Theta}) = \prod_k p(\boldsymbol{\Theta}_k)$, the GEM update given by $\mathcal{A}_{GEM}(p_m(\mathbf{X}|\boldsymbol{\Theta}), p_d(\mathbf{X}))$ is $p_m$-local.*

*Proof.* Rather than minimize $\mathrm{KL}[p_d(\mathbf{X}_o)||p_m(\mathbf{X}_o|\boldsymbol{\Theta})]$, the GEM algorithm minimizes an upper bound (the variational free energy). Taking $\mathbf{X} = [\mathbf{X}_o, \mathbf{X}_h]$:

$$\mathrm{KL}[p_d(\mathbf{X}_o)||p_m(\mathbf{X}_o|\boldsymbol{\Theta})] \geq \mathrm{KL}[p_d(\mathbf{X}_o)||p_m(\mathbf{X}_o|\boldsymbol{\Theta})] + \mathbb{E}_{p_d(\mathbf{X}_o)}\left[\mathrm{KL}[p_d(\mathbf{X}_h|\mathbf{X}_o)||p_m(\mathbf{X}_h|\mathbf{X}_o, \boldsymbol{\Theta})]\right]$$

$$= \mathrm{KL}[p_d(\mathbf{X})||p_m(\mathbf{X}|\boldsymbol{\Theta})], \tag{E.11}$$

where the inequality follows from the positivity of the KL divergence. Here, $p_d(\mathbf{X}_h|\mathbf{X})$ is an *approximate inference* distribution. Different choices of how this distribution is selected/optimized can produce very different learning algorithms, with varying degrees of biological plausibility. Obviously, the loss is minimized with respect to $p_d(\mathbf{X}_h|\mathbf{X})$ if $p_d(\mathbf{X}_h|\mathbf{X}) = p_m(\mathbf{X}_h|\mathbf{X},\boldsymbol{\Theta}_0)$ (where $\boldsymbol{\Theta}_0 = \boldsymbol{\Theta}$ *prior* to optimization wrt $\boldsymbol{\Theta}$). This choice corresponds to GEM [45]. For now, we will not concern ourselves with how $p_d(\mathbf{X}_h|\mathbf{X})$ is selected–instead, we will focus on the locality properties of gradient updates of this loss with respect to $\boldsymbol{\Theta}_m$.

Having packaged hidden and observed variables together ($\mathbf{X} = [\mathbf{X}_o, \mathbf{X}_h]$), our derivation proceeds exactly the same as for MLE:

$$
\begin{aligned}
\mathcal{A}_{GEM}(p_m(\mathbf{X}|\boldsymbol{\Theta}), p_d(\mathbf{X})) &\propto -\frac{\partial}{\partial\boldsymbol{\Theta}}\mathrm{KL}[p_d(\mathbf{X})||p_m(\mathbf{X}|\boldsymbol{\Theta})] \\
&= \frac{\partial}{\partial\boldsymbol{\Theta}}\int \log\left(\frac{p_m(\mathbf{X}|\boldsymbol{\Theta})}{p_d(\mathbf{X})}\right)p_d(\mathbf{X})d\mathbf{X} \\
&= \int \frac{\partial}{\partial\boldsymbol{\Theta}}\log\left(p_m(\mathbf{X}|\boldsymbol{\Theta})\right)p_d(\mathbf{X})d\mathbf{X} \\
&\approx \frac{1}{K}\sum_{k=0}^{K}\frac{\partial}{\partial\boldsymbol{\Theta}}\log\left(p_m(\mathbf{X}_h, \mathbf{X}|\boldsymbol{\Theta})\right),
\end{aligned}
\tag{E.12}
$$

where as with the MLE update, $\mathbf{X}_k \sim p_m(\mathbf{X}|\boldsymbol{\Theta})$. This update is $p_m$-local for the same reason that the MLE update is. $\qquad\square$

## E.4 Predictive Coding (PC)

As an additional note, if one takes the approximate posterior $p_d(\mathbf{X}_h|\mathbf{X}_o)$ to be given by:

$$
p_d(\mathbf{X}_h|\mathbf{X}_o) = \operatorname*{argmin}_{p_d(\mathbf{X}_h|\mathbf{X}_o)} \mathrm{KL}[p_d(\mathbf{X})||p_m(\mathbf{X}|\boldsymbol{\Theta})] \; s.t. \; p_d(\mathbf{X}_h|\mathbf{X}_o) \sim \delta(\bar{\mathbf{X}}_h(\mathbf{X}_o)),
\tag{E.13}
$$

where $\delta(\cdot)$ indicates a Dirac delta distribution and $\bar{\mathbf{X}}_h(\mathbf{X}_o)$ indicates a set of observation-dependent mean parameters, then we recover the predictive coding family of algorithms [35]. Typically, in this context for a given observed stimulus $\mathbf{X}_o$, $p_d(\mathbf{X}_h|\mathbf{X}_o) \sim \delta(\bar{\mathbf{X}}_h(\mathbf{X}_o))$ is estimated by reparameterization and gradient descent with respect to $\bar{\mathbf{X}}_h(\mathbf{X}_o)$ (a mean parameter that is observation-dependent), which—for clever choices of $p_m$—loosely resembles the dynamics of a recurrent neural network relaxing to a stimulus-conditioned equilibrium state [33]. After estimating $\bar{\mathbf{X}}_h(\mathbf{X}_o)$, parameters $\boldsymbol{\Theta}$ are updated as in GEM. Therefore, the derivation above also applies to predictive coding algorithms, which are consequently also $p_m$-local.

**Theorem E.4.** *If $p(\boldsymbol{\Theta}) = \prod_k p(\boldsymbol{\Theta}_k)$, the PC update given by $\mathcal{A}_{PC}(p_m(\mathbf{X}|\boldsymbol{\Theta}), p_d(\mathbf{X}))$ is $p_m$-local.*

## E.5 Wake-Sleep

Unlike the previous three examples, which only require sampling from the $p_d$ distribution and only calculate parameter updates according to the $p_m$ distribution, the Wake-Sleep algorithm parameterizes both distributions and jointly samples from a mixture of the two distributions across its 'wake' and 'sleep' phases. As we will see, this will mean that the Wake-Sleep algorithm will end up being $\gamma p_{md}$-local, where $\gamma$ is the binary variable that controls whether the system is in its 'wake' or 'sleep' phase.

**Theorem E.5.** *If $p(\boldsymbol{\Theta}, \boldsymbol{\Theta}^{(d)}) = \left(\prod_k p(\boldsymbol{\Theta}_k)\right)\left(\prod_k p(\boldsymbol{\Theta}_k^{(d)})\right)$, the Wake-Sleep estimator given by $\mathcal{A}_{WS}(p_m(\mathbf{X}|\boldsymbol{\Theta}), p_d(\mathbf{X}|\boldsymbol{\Theta}^{(d)}))$ is $\gamma p_{md}$-local, where $p_{md} = Mix\left(p_m(\mathbf{X}|\boldsymbol{\Theta}), p_d(\mathbf{X}|\boldsymbol{\Theta}^{(d)})\right)p(\boldsymbol{\Theta}, \boldsymbol{\Theta}^{(d)})$.*

*Proof.* Our updates use a similar loss to the GEM algorithm, namely we take:

$$
\mathcal{A}_{WS}(p_m(\mathbf{X}|\boldsymbol{\Theta}), p_d(\mathbf{X}|\boldsymbol{\Theta}^{(d)})) = \left[\Delta\boldsymbol{\Theta}_{WS}, \Delta\boldsymbol{\Theta}_{WS}^{(d)}\right],
\tag{E.14}
$$

where $\Delta\boldsymbol{\Theta}_{WS}$ is given by:

$$\Delta\boldsymbol{\Theta}_{WS} \propto -\frac{\partial}{\partial\boldsymbol{\Theta}}\mathrm{KL}[p_d(\mathbf{X}|\boldsymbol{\Theta}^{(d)})||p_m(\mathbf{X}|\boldsymbol{\Theta})]$$

$$\approx \frac{2}{K}\sum_{k=0}^{K}\gamma_k\frac{\partial}{\partial\boldsymbol{\Theta}}\log\left(p_m(\mathbf{X}_h,\mathbf{X}|\boldsymbol{\Theta})\right). \tag{E.15}$$

Here, $\gamma_k$, $\mathbf{X} \sim p_{md}(\mathbf{X},\gamma|\boldsymbol{\Theta},\boldsymbol{\Theta}^{(d)})$, whereas for GEM, we sampled only from $p_d$. Because each term of this update is 0 if $\gamma_k \neq 1$, this update is still an unbiased estimate of the gradient [46] and is effectively equivalent to the GEM update, except that it allows the system to alternate between sampling from $p_m$ and $p_d$. This alternation is useful because it will allow also calculating parameter updates for $\Delta\boldsymbol{\Theta}^{(d)}$. For $\boldsymbol{\Theta}^{(d)}$, we optimize the *reverse* KL-divergence[4]; by a directly analogous derivation to Eq. E.12, the update is given by:

$$\Delta\boldsymbol{\Theta}_{WS}^{(d)} \propto -\frac{\partial}{\partial\boldsymbol{\Theta}^{(d)}}\mathrm{KL}[p_m(\mathbf{X}|\boldsymbol{\Theta})||p_d(\mathbf{X}|\boldsymbol{\Theta}^{(d)})]$$

$$\approx \frac{2}{K}\sum_{k=0}^{K}(1-\gamma_k)\frac{\partial}{\partial\boldsymbol{\Theta}^{(d)}}\log\left(p_d(\mathbf{X}|\boldsymbol{\Theta}^{(d)})\right). \tag{E.16}$$

Now, these updates contain the score function for both $p_m$ and $p_d$, as well as the scalar mixture variable $\gamma$. As a consequence, both updates are $\gamma p_{md}$-local, via Properties 2.7 and 2.8 (slightly more precisely, $\Delta\boldsymbol{\Theta}$ is $\gamma p_m$-local, and $\Delta\boldsymbol{\Theta}^{(d)}$ is $\gamma p_d$-local). $\qquad\square$

### E.6    Impression Learning (IL)

The impression learning parameter update [23] is closely related to the Wake-Sleep parameter update, and is consequently also $\gamma p_{md}$-local. What distinguishes IL from WS is the use of rapid alternations in the gating signal $\gamma_t$ within a single trial with $T$ time steps. Here, $\mathbf{X} = [\mathbf{X}_0,...,\mathbf{X}_T]$, $\gamma = [\gamma_0,...,\gamma_t]$ and $p_{md}(\mathbf{X}|\gamma,\boldsymbol{\Theta}) = \prod_{t=0}^{T}p_d(\mathbf{X}_t|\mathbf{X}_{t-1},\boldsymbol{\Theta})^{\gamma_t}p_m(\mathbf{X}_t|\mathbf{X}_{t-1},\boldsymbol{\Theta})^{1-\gamma}$ is a mixture distribution in which $\gamma_t$ alternates between 0 and 1, sampling from either $p_m$ or $p_d$ at the time step $t$, respectively.

**Theorem E.6.** *If* $p(\boldsymbol{\Theta}) = \prod_k p(\boldsymbol{\Theta}_k)$, *the impression learning estimator given by* $\mathcal{A}_{IL}(p_m(\mathbf{X}|\boldsymbol{\Theta}),p_d(\mathbf{X}|\boldsymbol{\Theta}))$ *is* $\gamma p_{md}$-*local, where* $p_{md} = Mix\left(p_m(\mathbf{X}|\boldsymbol{\Theta}),p_d(\mathbf{X}|\boldsymbol{\Theta}^{(d)})\right)p(\boldsymbol{\Theta})$.

*Proof.* Similar to WS, the update is given by:

$$\Delta\boldsymbol{\Theta}_{IL} \propto \int\left[\sum_{t=0}^{T}\frac{\partial}{\partial\boldsymbol{\Theta}}[(1-\gamma_t)\log p_d(\mathbf{X}_t|\mathbf{X}_{t-1},\boldsymbol{\Theta}) + \gamma_t p_m(\mathbf{X}_t|\mathbf{X}_{t-1},\boldsymbol{\Theta})]\right]p_{md}(\mathbf{X}|\gamma,\boldsymbol{\Theta})d\mathbf{X} \tag{E.17}$$

$$\approx \sum_{t=0}^{T}(1-\gamma_t)\frac{\partial}{\partial\boldsymbol{\Theta}}\log p_d(\mathbf{X}_t|\mathbf{X}_{t-1},\boldsymbol{\Theta}) + \gamma_t\frac{\partial}{\partial\boldsymbol{\Theta}}p_m(\mathbf{X}_t|\mathbf{X}_{t-1},\boldsymbol{\Theta}), \tag{E.18}$$

where in this last equality, $\mathbf{X} \sim p_{md}(\mathbf{X})$, and we are performing a single-sample gradient approximation. It is worth noting that unlike in the Wake-Sleep algorithm, here $\gamma_t$ is not a constant throughout time. Instead, $\gamma_t$ alternates between 0 and 1 with 'phase duration' $K$, i.e. $\gamma_{t+1} = 1 - \gamma_t$ if $\mathrm{mod}\,(t,K) = 0$, and $\gamma_{t+1} = \gamma_t$ otherwise. The IL update is the score function of $p_{md}(\mathbf{X}|1-\gamma,\boldsymbol{\Theta})$, which has identical dependencies to the score function of $p_{md}(\mathbf{X}|\gamma,\boldsymbol{\Theta})$ (only a change from $\gamma \to 1-\gamma$ has occurred). Therefore, if $p(\boldsymbol{\Theta}) = \prod_k p(\boldsymbol{\Theta}_k)$, this parameter update is $\gamma p_{md}$-local by Property 2.7. $\qquad\square$

---

[4]A rigorous discussion of why this optimization process is sensible is beyond the scope of this manuscript. See [22, 47] for more detail.

## E.7 Contrastive Divergence for Boltzmann machines (CD)

While the GEM learning update is provably $p_m$-local, it is also predicated on the assumption that the parameter marginal distribution factorizes as $p(\mathbf{\Theta}) = \prod_k p(\mathbf{\Theta}_k)$, which as we note in Appendix D can be difficult to ensure for even simple undirected graphical models. An extension of the GEM algorithm, we can show that the CD algorithm is $\gamma p_{md}$-local (as opposed to just $p_m$-local) under less restrictive assumptions. The cost of this is that CD learning usually requires costly MCMC sampling from both the posterior distribution $p_m(\mathbf{X}_h | \mathbf{X}_o, \mathbf{\Theta})$ and the full joint distribution $p_m(\mathbf{X}_h, \mathbf{X}_o | \mathbf{\Theta})$.

**Theorem E.7.** *The CD update given by $\mathcal{A}_{CD}(p_m(\mathbf{X}|\mathbf{\Theta}), p_d(\mathbf{X}))$ is $\gamma p_{md}$-local, where $p_{md} = Mix\left(p_m(\mathbf{X}|\mathbf{\Theta}), p_d(\mathbf{X})\right) p(\mathbf{\Theta})$.*

As mentioned above, for GEM the most natural choice for $p_d(\mathbf{X}_h|\mathbf{X})$ is given by $p_m(\mathbf{X}_h|\mathbf{X}, \mathbf{\Theta}_0)$. As we demonstrated, parameter updates calculated according to this rule will be $\gamma p_{md}$-local, but there are two important caveats. First, for GEM to produce biologically plausible updates, we still need a biological system that can sample from $p_m(\mathbf{X}_h|\mathbf{X}, \mathbf{\Theta}_0)$. Second, it is important to remember that we are only guaranteed that the score function is guaranteed to be $p$-local if the marginal parameter probability distribution factorizes as $p(\mathbf{\Theta}) = \prod_k p(\mathbf{\Theta}_k)$. For a DAG, it may be difficult to satisfy the first condition without approximation (given by the Wake-Sleep algorithm, for instance), whereas for an UG, it may be difficult to satisfy the second condition, as we saw in Section D. To make our update $\gamma p_{md}$-local for an undirected graphical model like the Boltzmann machine, we will require an extra step that we outline here to use the energy function Property 2.6 rather than the score function Property 2.7.

Having committed to working with an undirected graphical model, instead of sticking to the original GEM update, here we break apart the probability distribution as:

$$\mathcal{A}_{CD}(p_m(\mathbf{X}|\mathbf{\Theta}), p_d(\mathbf{X})) = \int \frac{\partial}{\partial \mathbf{\Theta}} \log\left(p_m(\mathbf{X}|\mathbf{\Theta})\right) p_d(\mathbf{X}) d\mathbf{X} \tag{E.19}$$

$$= \int \frac{\partial}{\partial \mathbf{\Theta}} \left[\log\left(E(\mathbf{X}, \mathbf{\Theta})\right) - \log \mathcal{Z}(\mathbf{\Theta})\right] p_d(\mathbf{X}) d\mathbf{X} \tag{E.20}$$

$$= -\int \frac{\partial}{\partial \mathbf{\Theta}} E(\mathbf{X}, \mathbf{\Theta}) p_d(\mathbf{X}) d\mathbf{X} - \frac{\partial}{\partial \mathbf{\Theta}} \log \mathcal{Z}(\mathbf{\Theta}) \tag{E.21}$$

$$= -\int \frac{\partial}{\partial \mathbf{\Theta}} E(\mathbf{X}, \mathbf{\Theta}) p_d(\mathbf{X}) d\mathbf{X} - \frac{1}{\mathcal{Z}(\mathbf{\Theta})} \int \frac{\partial}{\partial \mathbf{\Theta}} e^{-\frac{E(\mathbf{X}, \mathbf{\Theta})}{\sigma^2}} d\mathbf{X} \tag{E.22}$$

$$= -\int \frac{\partial}{\partial \mathbf{\Theta}} E(\mathbf{X}, \mathbf{\Theta}) p_d(\mathbf{X}) d\mathbf{X} + \frac{1}{\mathcal{Z}(\mathbf{\Theta})} \int \frac{\partial}{\partial \mathbf{\Theta}} E(\mathbf{X}, \mathbf{\Theta}) e^{-E(\mathbf{X}, \mathbf{\Theta})} d\mathbf{X} \tag{E.23}$$

$$= -\int \frac{\partial}{\partial \mathbf{\Theta}} E(\mathbf{X}, \mathbf{\Theta}) p_d(\mathbf{X}) d\mathbf{X} + \int \frac{\partial}{\partial \mathbf{\Theta}} E(\mathbf{X}, \mathbf{\Theta}) p_m(\mathbf{X}|\mathbf{\Theta}) d\mathbf{X} \tag{E.24}$$

$$\approx \frac{2}{K} \sum_{k=0}^{K} (-1)^{\gamma_k} \frac{\partial}{\partial \mathbf{\Theta}} E(\mathbf{X}_k, \mathbf{\Theta}), \tag{E.25}$$

where $\mathbf{X}_k$ is sampled from $p_{md}$. Now, by Property 2.6 and Property 2.8, this update is $\gamma p_{md}$-local. This is the Boltzmann machine learning algorithm [26], where the clamped and unclamped phases are alternated between stochastically [46]; for our previous linear Boltzmann example, in which $\mathbf{X} = \mathbf{r}$ and $\mathbf{\Theta} = \mathbf{W}$, the derivative of the energy function (Eq. D.3) with respect to a parameter $\mathbf{W}_{ij}$ is $\mathbf{r}_i \mathbf{r}_j$, demonstrating that updates correspond to two phases of updates: one in which $\mathbf{r}_o$ is clamped to a data distribution given by $p_d(\mathbf{r}_o)$ for some $\mathbf{r}_o \subseteq \mathbf{r}$ and Hebbian updates are positive, contrasted with an unclamped phase in which updates are negative. Note that for this model approximate sampling from a posterior distribution $p_m(\mathbf{r}_{\neq o}|\mathbf{r}_o, \mathbf{W}) p_d(\mathbf{r}_0)$ is no more difficult than sampling from the joint distribution: one simply holds $\mathbf{r}_o$ fixed to an environmental data sample and performs Langevin sampling on all other variables.

### E.8 Equilibrium Propagation (EP)

Though the derivation and setup of the equilibrium propagation algorithm [27] is very different from Contrastive Divergence, the functional form of the derived update is very similar. While equilibrium propagation typically operates on deterministic networks, here we will provide our derivation for the stochastic version with an energy function defining a joint distribution over $\Theta$ and $\mathbf{X}$ (as in Section D), which is somewhat more straightforward to fit into the $p$-locality framework.

Suppose that we have a probabilistic energy-based model whose energy function is given by:

$$E(\mathbf{Z}, \epsilon) = E_0(\mathbf{Z}) + \epsilon \mathcal{L}(\mathbf{Z}), \tag{E.26}$$

where $\epsilon \in \{0, \beta\}$, where $\beta \ll 1$, where $\mathcal{L}(\mathbf{Z})$ is the loss function that the parameter updates are optimizing. This can be thought of as a 'soft-clamped' system, in which nonzero $\epsilon$ pushes the system towards slightly better performance. Intuitively, the EP parameter update attempts to change network dynamics so that the unclamped system is nudged towards the slightly better performing soft-clamped system. Then we have the following theorem:

**Theorem E.8.** *The EP update given by* $\mathcal{A}_{EP}(p(\mathbf{X}|\Theta, \gamma))$ *is* $\gamma p_{md}$-*local, where* $p_{md} = Mix\left(p(\mathbf{X}|\Theta, \epsilon = \beta), p(\mathbf{X}|\Theta, \epsilon = 0)\right) p(\Theta)$.

The parameter update for equilibrium propagation is given by:

$$\Delta\Theta_{EP} \propto -\frac{1}{\beta} \left[ \mathbb{E}_{\epsilon=\beta} \frac{\partial E(\mathbf{Z}, \epsilon)}{\partial \Theta} - \mathbb{E}_{\epsilon=0} \frac{\partial E(\mathbf{Z}, \epsilon)}{\partial \Theta} \right] \tag{E.27}$$

$$\approx \frac{2}{\beta} \sum_{k=0}^{K} (-1)^{\epsilon_k/\beta} \frac{\partial E(\mathbf{Z}_k, \epsilon_k)}{\partial \Theta}, \tag{E.28}$$

where for the final equality we are using a sampling-based approximation in which we are sampling from $\mathbf{X}_k \sim p(\mathbf{X}|\Theta, \epsilon_k)$, and $\epsilon_k/\beta \sim Bernoulli(0.5)$. This is almost identical to the Contrastive Divergence update, except that rather than clamping neural activities to a target output, they are slightly biased towards better performance. Because this is the combination of the derivative of the energy function with a mixture variable $\gamma = \epsilon_k/\beta$, by Properties 2.6 and 2.8, this update is $\gamma p_{md}$-local where $p_{md} = Mix\left(p(\mathbf{X}|\Theta, \epsilon = \beta), p(\mathbf{X}|\Theta, \epsilon = 0)\right) p(\Theta)$.

### E.9 Winner-take-all STDP

While Contrastive Divergence uses MCMC sampling to approximate the GEM update, Nessler et al. [48] use a particular generative model for which the posterior can be analytically calculated and resembles a simple winner-take-all neural circuit. Then, the authors derive their STDP parameter update as an approximation to the GEM algorithm. Because of this, one might imagine that the derived STDP update may, like the GEM algorithm, be $p_m$-local. We will see below that this is the case.

First, we define the generative model $p_m$ used in the paper:

$$p_m(\mathbf{r}, \mathbf{s}|\mathbf{W}) = \frac{1}{\mathcal{Z}} e^{-E(\mathbf{r}, \mathbf{s}, \mathbf{W})} \tag{E.29}$$

$$E(\mathbf{r}, \mathbf{s}, \mathbf{W}) = -\left( \sum_{i=0}^{N} \mathbf{r}_i \mathbf{W}_{i0} + \sum_{i=0}^{N} \sum_{j=0}^{N_s} \mathbf{r}_i \mathbf{W}_{ij} \mathbf{s}_j \right), \tag{E.30}$$

where $\mathcal{Z}$ is the normalizing constant and the $\mathbf{r}$ and $\mathbf{s}$ vectors contain binary random variables. Furthermore, in the network only one neuron $\mathbf{r}$ is assumed to fire at any given time ($\mathbf{r}_i \neq 0 \Leftrightarrow \mathbf{r}_k = 0 \ \forall k \neq i$). The inference distribution, conditioned on a stimulus $\mathbf{s}$ can be calculated as follows:

$$p_m(\mathbf{r}|\mathbf{s}, \mathbf{W}) = \frac{\exp\left( \sum_{i=0}^{N} \mathbf{r}_i \mathbf{W}_{i0} + \sum_{i=0}^{N} \sum_{j=0}^{N_s} \mathbf{r}_i \mathbf{W}_{ij} \mathbf{s}_j \right)}{\sum_{i=0}^{N} \exp\left( \mathbf{W}_{i0} + \sum_{j=0}^{N_s} \mathbf{W}_{ij} \mathbf{s}_j \right)}. \tag{E.31}$$

This probability distribution can be interpreted as a kind of winner-take-all computation, dominated by the neuron with the greatest input current [48]. Samples from this distribution are used to compute the weight updates:

$$\Delta \mathbf{W}_{ij} \propto \mathbf{r}_i \left( c\mathbf{s}_i e^{-\mathbf{W}_{ij}} - 1 \right) \tag{E.32}$$

$$\Delta \mathbf{W}_{i0} \propto \mathbf{r}_i e^{-\mathbf{W}_{i0}} - 1, \tag{E.33}$$

where $c$ is a positive constant. We now prove the following theorem assessing the $p$-locality of this distribution:

**Theorem E.9.** *The STDP update given by $\mathcal{A}_{STDP}(p(\mathbf{X}|\boldsymbol{\Theta}))$ is $\gamma p_m$-local.*

*Proof.* To see this, we first note that the gradient of the energy function $E(\mathbf{r}, \mathbf{s}, \mathbf{W})$ with respect to the parameters is $p_m$-local by Property 2.6. Therefore, any variables contained within this will also be permissible under $p_m$-locality.

For $\mathbf{W}_{ij}$, we have:

$$\frac{\partial E(\mathbf{r}, \mathbf{s}, \mathbf{W})}{\partial \mathbf{W}_{ij}} = -\mathbf{r}_i \mathbf{s}_j, \tag{E.34}$$

so that we know $\mathbf{r}_i$ and $\mathbf{s}_j$ are permissible for $\Delta \mathbf{W}_{ij}$ under $p_m$-locality; further, the value of a parameter itself, $\mathbf{W}_{ij}$, is always allowed under $p_m$-locality. These are the only variables on which $\Delta \mathbf{W}_{ij}$ depends, so this update is $p_m$-local.

For $\mathbf{W}_{i0}$, we have:

$$\frac{\partial E(\mathbf{r}, \mathbf{s}, \mathbf{W})}{\partial \mathbf{W}_{i0}} = -\mathbf{r}_i, \tag{E.35}$$

so that we know $\mathbf{r}_i$ is permissible for $\Delta \mathbf{W}_{i0}$. By the same reasoning, this update is also $p_m$-local. Since all updates are therefore $p_m$-local, we may conclude that the full algorithm is $p_m$-local. However, this proof does not have the same level of generality as for the previous algorithms, because the algorithm is only defined for a single winner-take-all network model. $\square$

### E.10 Backpropagation

**Theorem E.10.** *If $p(\boldsymbol{\Theta}) = \prod_k p(\boldsymbol{\Theta}_k)$ and $p(\mathbf{X}|\boldsymbol{\Theta})$ is defined by Eq. 3 (with $\mathbf{X} = \mathbf{r}$ and $\boldsymbol{\Theta} = \mathbf{W}$), the BP update for $\mathbf{W}_{ij}^{(l)}$ with a loss $\mathcal{L}(\mathbf{r})$, given by $\mathcal{A}_{BP}(p(\mathbf{r}|\mathbf{W}), \mathcal{L}(\mathbf{r}))$ is $\mathbf{e}_i^{(l)}p$-local, where $\mathbf{e}_i^{(l)} = \frac{\mathrm{d}L}{\mathrm{d}\bar{\mathbf{r}}_i}$. Similarly, the updates for feedback alignment, weight mirror, and Burstprop are $\hat{\mathbf{e}}_i^{(l)}p$-local, where $\hat{\mathbf{e}}_i^{(l)}$ is given by their respective gradient approximations.*

*Proof.* As a first step, for clarity purposes we will demonstrate that backpropagation [28], is *not* $p$-local with respect to the simple feedforward neural network architecture we outlined above; we will subsequently demonstrate that it and its approximations do satisfy a particular notion of $\mathbf{S}p$-locality. For a scalar loss function $\mathcal{L}(r^L)$ and a single parameter $\mathbf{W}_{ij}^{(l)}$, the backpropagation gradient is given by the negative gradient of the loss with respect to the parameter of choice, using the reparameterization trick [25, 24] to take for a single sample of $\{\mathbf{r}^{(l)}\}_{l=0:L}$ the mean (noiseless) mapping from $\mathbf{r}^{(l-1)} \to \mathbf{r}^{(l)}$ to be $\bar{\mathbf{r}}^{(l)} = h(\mathbf{W}^{(l)}\mathbf{r}^{(l-1)})$, so that by Eq. 5, we have $\mathbf{r}^{(l)} = \bar{\mathbf{r}}^{(l)} + \sigma\boldsymbol{\eta}^{(l)}$. By the chain rule, gradient descent gives:

$$\Delta \mathbf{W}_{ij}^{(l)} \propto -\frac{\mathrm{d}\mathcal{L}(\mathbf{r}^{(L)})}{\mathrm{d}\bar{\mathbf{r}}^{(L)}} \left( \prod_{k=l}^{L} \frac{\mathrm{d}\bar{\mathbf{r}}^{(k)}(\mathbf{r}^{(k-1)})}{\mathrm{d}\bar{\mathbf{r}}^{(k-1)}} \right) \frac{\mathrm{d}\bar{\mathbf{r}}^{(l)}(\mathbf{W}_{ij}^{(l)})}{\mathrm{d}\mathbf{W}_{ij}^{(l)}}. \tag{E.36}$$

Based on our analysis in Section 2.4, this update function is clearly not $p$-local, because the update depends on firing rates $\mathbf{r}^{(k)}$ for $k > l$. However, while backpropagation is not in general $p$-local, any algorithm can be $\mathbf{S}p$-local: for example, if we take $\mathbf{S} = \mathbf{Z}$, then by Definition 2.3, any parameter update can contain any variable in the graphical model $p(\mathbf{Z})$. Taking $\mathbf{S} = \mathbf{Z}$ is inherently vacuous: $\mathbf{S}p$-locality is only conceptually useful if we can cleanly reduce the number of variables included in $\mathbf{S}$ for a broad set of biologically relevant neural architectures. Fortunately, for backpropagation

operating on a feedforward neural network governed by Equation 5, we do not need to include every variable in the network. By Equation E.36, we see for our example feedforward neural network that:

$$\Delta \mathbf{W}_{ij}^{(l)} \propto -\frac{\mathrm{d}\mathcal{L}}{\mathrm{d}\bar{\mathbf{r}}_i^{(l)}} \frac{\mathrm{d}\bar{\mathbf{r}}_i^{(l)}}{\mathrm{d}\mathbf{W}_{ij}^{(l)}}, \tag{E.37}$$

where $\frac{\mathrm{d}\mathcal{L}}{\mathrm{d}\bar{\mathbf{r}}_i^{(l)}} = \frac{\mathrm{d}\mathcal{L}(\mathbf{r}^{(L)})}{\mathrm{d}\bar{\mathbf{r}}^{(L)}} \left( \prod_{k=l+2}^{L} \frac{\mathrm{d}\bar{\mathbf{r}}^{(k)}(\mathbf{r}^{(k-1)})}{\mathrm{d}\bar{\mathbf{r}}^{(k-1)}} \right) \frac{\mathrm{d}\bar{\mathbf{r}}^{(l+1)}(\mathbf{r}^{(l)})}{\mathrm{d}\bar{\mathbf{r}}_i^{(l)}}$ is the derivative of the global loss function with respect to the individual mean neuron activation $\bar{\mathbf{r}}_i^{(l)}$. Interestingly, $\frac{\mathrm{d}\bar{\mathbf{r}}_i^{(l)}}{\mathrm{d}\mathbf{W}_{ij}^{(l)}}$—the derivative of the mean parameter for neuron $\mathbf{r}_i$—is a function of only the parents of $\mathbf{r}^{(l)}$, which are therefore the *coparents* of $\mathbf{W}_{ij}^{(l)}$. To verify that this particular component of the weight update is $p$-local, we can compare its dependencies to the score function, which is in this case $p$-local by Property 2.7. As noted in Section 2.4, the score function is given by:

$$\frac{\partial \log p(\mathbf{r}|\mathbf{s}, \mathbf{W})}{\partial \mathbf{W}_{ij}^{(l)}} = \frac{\left( \mathbf{r}_i^{(l)} - h(\mathbf{V}_i^{(l)}) \right)}{\sigma^2} h'(\mathbf{V}_i^{(l)}) \mathbf{r}_j^{(l-1)}, \tag{E.38}$$

where $\mathbf{V}_i^{(l)} = \mathbf{W}_{i:}^{(l)} \mathbf{r}^{(l-1)}$. Because the score function is $p$-local, any variables that it depends on are permissible for $p$-local updates. The score function depends on $\mathbf{r}_i^{(l)}$ and $\mathbf{r}^{(l-1)}$, whereas $\frac{\mathrm{d}\bar{\mathbf{r}}_i^{(l)}}{\mathrm{d}\mathbf{W}_{ij}^{(l)}}$ depends only on $\mathbf{r}^{(l-1)}$. It follows that this function is also $p$-local.

As we already discussed, $\frac{\mathrm{d}\mathcal{L}}{\mathrm{d}\bar{\mathbf{r}}_i^{(l)}}$ is not $p$-local because it depends on neurons downstream of $\mathbf{r}_i^{(l)}$. However, if we define an auxiliary random variable $\mathbf{e}_i^{(l)} = \frac{\mathrm{d}\mathcal{L}}{\mathrm{d}\bar{\mathbf{r}}_i^{(l)}}$, we see that because it multiplies $\mathbf{e}_i^{(l)}$ with a $p$-local function, $\mathbf{W}_{ij}^{(l)}$ is $\mathbf{e}_i^{(l)} p$-local.

Importantly, this does not mean that backpropagation is biologically plausible: this notion of locality provides no clues as to how $\mathbf{e}_i^{(l)}$ could be calculated or approximated in the brain, and an explicit calculation of gradients could not be possible due to the weight transport problem [20]. There are many recent models that account for how $\mathbf{e}_i^{(l)}$ could be approximated by an approximate credit assignment signal $\hat{\mathbf{e}}_i^{(l)}$ involving either random feedback synapses that project errors backwards through the network (feedback alignment [29]) or feedback synapses that dynamically adjust through local synaptic mechanisms so that $\hat{\mathbf{e}}_i^{(l)}$ provides an unbiased approximation (e.g. weight mirror or Kolen-Pollack alignment [30], and BurstProp [31]). Each of these algorithms decomposes into a nonlocal feedback term $\hat{\mathbf{e}}_i^{(l)}$ and a $p$-local term in exactly the same way, and are consequently $\hat{\mathbf{e}}_i^{(l)} p$-local. □

### E.11    Real Time Recurrent Learning (RTRL)

Consider an autonomous recurrent neural network whose directed acyclic graphical model is provided by the following equations (we will ignore stimulus-dependence for notational simplicity):

$$p(\mathbf{r}|\mathbf{W}) = p(\mathbf{r}(0)) \prod_{t=1}^{T} \prod_{i=1}^{N} p(\mathbf{r}_i(t)|\mathbf{r}(t-1), \mathbf{W}) \tag{E.39}$$

$$p(\mathbf{r}_i(t)|\mathbf{r}_i(t-1), \mathbf{W}) \sim \mathcal{N}(h(\mathbf{W}_{i:}\mathbf{r}(t-1)), \sigma^2), \tag{E.40}$$

where $p(\mathbf{r}(0))$ corresponds to some initial distribution of activity states. This probability distribution of firing rates corresponds to the following neural sampling dynamics:

$$\mathbf{r}(t) = h(\mathbf{W}\mathbf{r}(t-1)) + \sigma\boldsymbol{\eta}, \tag{E.41}$$

where $\boldsymbol{\eta} \sim \mathcal{N}(0,1)$. For this model, we have the following theorem:

**Theorem E.11.** *If $p(\boldsymbol{\Theta}) = \prod_k p(\boldsymbol{\Theta}_k)$ and $p(\mathbf{X}|\boldsymbol{\Theta})$ is defined by Eq. E.39 (with $\mathbf{X} = \mathbf{r}$ and $\boldsymbol{\Theta} = \mathbf{W}$), the RTRL update for $\mathbf{W}_{ij}$ with a loss $\mathcal{L}(\mathbf{r}(T))$, given by $\mathcal{A}_{RTRL}(p(\mathbf{r}|\mathbf{W}), \mathcal{L}(\mathbf{r}))$ is $\mathbf{eJ}p$-local, where $\mathbf{e} = \frac{\partial L(\mathbf{r}(T))}{\partial \bar{\mathbf{r}}(T)}$, and $\mathbf{J} = \{\mathbf{J}(t) = \frac{\partial \bar{\mathbf{r}}(t,\mathbf{r})}{\partial \bar{\mathbf{r}}(t-1,\mathbf{r})}\}$.*

*Proof.* The directed graphical model corresponding to these dynamics is depicted in Figure 1b: as with backpropagation, we will use the score function for our graphical model to identify permissible variables. For a single synapse, the score function is given by:

$$\frac{\partial \log p(\mathbf{r}|\mathbf{W})}{\partial \mathbf{W}_{ij}} = \sum_{t=1}^{T} \frac{\partial \log p(\mathbf{r}_i(t)|\mathbf{r}(t-1), \mathbf{W})}{\partial \mathbf{W}_{ij}} \tag{E.42}$$

$$= \sum_{t=1}^{T} \frac{(\mathbf{r}_i(t) - h(\mathbf{V}_i(t)))}{\sigma^2} h'(\mathbf{V}_i(t))\mathbf{r}_j(t-1), \tag{E.43}$$

where $\mathbf{V}_i(t) = \mathbf{W}_{i:}\mathbf{r}(t-1)$. Thus $p$-local parameter updates for $\mathbf{W}_{ij}$ may include $\mathbf{W}_{i:}$, $\mathbf{r}_i(t)$ and $\{\mathbf{r}_k(t-1) : \mathbf{W}_{ik} \neq 0\}$ $\forall t$. We will now compare these allowed variables to the RTRL update. As with backpropagation, we take $\bar{\mathbf{r}}(t, \mathbf{r}) = h(\mathbf{W}\mathbf{r}(t-1))$, so that $\mathbf{r}(t) = \bar{\mathbf{r}}(t, \mathbf{r}) + \sigma\boldsymbol{\eta}$. The RTRL update minimizes a loss $\mathcal{L}(\mathbf{r}(T))$ via the chain rule [49, 21]:

$$\Delta \mathbf{W}_{ij} \propto \frac{\partial \mathcal{L}(\mathbf{r}(T))}{\partial \bar{\mathbf{r}}(T)} \frac{\partial \bar{\mathbf{r}}(T, \mathbf{r})}{\partial \mathbf{W}_{ij}} \tag{E.44}$$

$$\frac{\partial \bar{\mathbf{r}}(t, \mathbf{r})}{\partial \mathbf{W}_{ij}} = \frac{\partial \bar{\mathbf{r}}(t, \mathbf{r})}{\partial \bar{\mathbf{r}}(t-1, \mathbf{r})} \frac{\partial \bar{\mathbf{r}}(t-1, \mathbf{r})}{\partial \mathbf{W}_{ij}} + g(\mathbf{r}(t-1)) \tag{E.45}$$

$$g(\mathbf{r}(t-1))_k = \begin{cases} h'(\mathbf{V}_i(t))\mathbf{r}_j(t-1) & \text{if } i = k \\ 0 & \text{otherwise.} \end{cases} \tag{E.46}$$

this second equation provides a recursive update equation which can be stored online as a trial progresses. The $g(\mathbf{r}(t-1))$ term is $p$-local, because it appears in Eq. E.43. However, $\frac{\partial \bar{\mathbf{r}}(t, \mathbf{r})}{\partial \bar{\mathbf{r}}(t-1, \mathbf{r})}$, an $N \times N$ Jacobian matrix, is not $p$-local, since it depends on all neurons in the network $\mathbf{r}(t-1)$ as well as all parameters $\mathbf{W}$—neurons that do not directly synapse onto neuron $\mathbf{r}_i$ and weights $\mathbf{W}_{kl}$ for $k \neq i$ are excluded from $p$-local updates by Property 2.1 according to the DAG defined by Eq. E.39. Furthermore, as we have seen with backpropagation, in general the credit assignment signal $\frac{\partial \mathcal{L}(\mathbf{r}(T))}{\partial \bar{\mathbf{r}}(T)}$ is not $p$-local. Therefore, to characterize the $\mathbf{S}p$-locality of RTRL, we will have to proceed similarly to backpropagation, and define auxiliary variables to include in the set $\mathbf{S}$.

As with backpropagation, we define the auxiliary random variable $\mathbf{e} = \frac{\partial \mathcal{L}(\mathbf{r}(T))}{\partial \bar{\mathbf{r}}(T)}$. Because we have found the Jacobians to also violate $p$-locality, we will also define the set of auxiliary variables $\mathbf{J} = \{\mathbf{J}(t) = \frac{\partial \bar{\mathbf{r}}(t, \mathbf{r})}{\partial \bar{\mathbf{r}}(t-1, \mathbf{r})}\}$. With these auxiliary variables, we can see that $\frac{\partial \bar{\mathbf{r}}(t, \mathbf{r})}{\partial \mathbf{W}_{ij}}$ is $\mathbf{J}p$-local $\forall t$, and consequently, the RTRL update is $\mathbf{eJ}p$-local. $\qquad \square$

This is, of course, not biologically plausible in any way. The set $\mathbf{J}$ allows the parameters to have access to the state of the entire network, at all time points, even from neurons that do not have any direct connections to the neuron whose synapse is being updated. Further, the entire error vector $\mathbf{e}$ is required to compute the update. This is even less plausible than backpropagation, which only required access to $\mathbf{e}_i$. However, the RTRL update is an important baseline for analyzing the locality properties of other learning algorithms that are constructed as approximations of it, namely e-prop and RFLO.

## E.12 e-prop

**Theorem E.12.** *If $p(\boldsymbol{\Theta}) = \prod_k p(\boldsymbol{\Theta}_k)$ and $p(\mathbf{X}|\boldsymbol{\Theta})$ is defined by Eq. E.39 (with $\mathbf{X} = \mathbf{r}$ and $\boldsymbol{\Theta} = \mathbf{W}$), the e-prop update for $\mathbf{W}_{ij}$ with a loss $\mathcal{L}(\mathbf{r}(T))$, given by $\mathcal{A}_{ep}(p(\mathbf{r}|\mathbf{W}), \mathcal{L}(\mathbf{r}))$ is $\mathbf{e}_i p$-local, where $\mathbf{e}_i = \frac{\partial L(\mathbf{r}(T)}{\partial \bar{\mathbf{r}}_i(T)}$.*

*Proof.* For e-prop [37], we will also consider networks constructed according to Eq. E.39. The update is almost identical to the RTRL update, but several terms will be discarded, allowing the update to be $\mathbf{e}_i p$-local, as opposed to the $\mathbf{eJ}p$-local update given by RTRL. The update is as follows:

$$\Delta \mathbf{W}_{ij} \propto \frac{\partial \mathcal{L}(\mathbf{r}(T))}{\partial \bar{\mathbf{r}}_i(T)} \frac{\partial \tilde{\mathbf{r}}_i(T, \mathbf{r})}{\partial \mathbf{W}_{ij}} \tag{E.47}$$

$$\frac{\partial \tilde{\mathbf{r}}_i(t, \mathbf{r})}{\partial \mathbf{W}_{ij}} = h'(\mathbf{V}_i(t))\mathbf{W}_{ii} \frac{\partial \tilde{\mathbf{r}}_i(t-1, \mathbf{r})}{\partial \mathbf{W}_{ij}} + h'(\mathbf{V}_i(t))\mathbf{r}_j(t-1). \tag{E.48}$$

This update combines the neuron-specific credit assignment signal $\mathbf{e}_i$ with a local 'eligibility trace' $\frac{\partial \tilde{\mathbf{r}}_i(t, \mathbf{r})}{\partial \mathbf{W}_{ij}}$ which performs approximate credit assignment by filtering and summing coactivity between neuron $i$ and neuron $j$ across timesteps. It is worth noting that the particular functional form of this eligibility trace is determined by our simplified RNN dynamics (Eq. E.39), which causes coactivity from previous timesteps to decay exponentially in proportion to the magnitude of the autapse $\mathbf{W}_{ii}$—alternative neural network dynamics using continuous-time dynamics, or adaptive neural firing thresholds may alter the functional form of the eligibility trace [37], but do not fundamentally alter the $p$-locality properties of the update. Now, we only need to show that the eligibility trace is $p$-local.

As with RTRL, we can observe that $h'(\mathbf{V}_i(t))$ and $\mathbf{r}_j(t-1)$ both appear in the score function for our RNN (Eq. E.43) for all timesteps, as does $\mathbf{W}_{ii} \subset \mathbf{W}_{i:}$. Because the score function is $p$-local, we know that these variables are all allowed under $p$-locality. The eligibility trace only depends on these terms, from both the current time step and, recursively, from previous timesteps. Therefore, the eligibility trace is $p$-local. The e-prop update is a multiplication between $\mathbf{e}_i$ and the eligibility trace, so by Def. 2.3 the update is $\mathbf{e}_i p$-local.

$\square$

## E.13    Random feedback local online learning (RFLO)

The RFLO update [36] is nearly identical to the e-prop update, except we replace $\mathbf{e}_i$ with an approximate credit assignment signal $\hat{\mathbf{e}}_i$ (which replaces symmetric feedback weights with random connections, similar to Feedback Alignment).

The update is given by:

$$\Delta \mathbf{W}_{ij} \propto \hat{\mathbf{e}}_i \frac{\partial \tilde{\mathbf{r}}_i(T, \mathbf{r})}{\partial \mathbf{W}_{ij}} \tag{E.49}$$

$$\frac{\partial \tilde{\mathbf{r}}_i(t, \mathbf{r})}{\partial \mathbf{W}_{ij}} = h'(\mathbf{V}_i(t))\mathbf{W}_{ii} \frac{\partial \tilde{\mathbf{r}}_i(t-1, \mathbf{r})}{\partial \mathbf{W}_{ij}} + h'(\mathbf{V}_i(t))\mathbf{r}_j(t-1). \tag{E.50}$$

Following exactly the same reasoning as with e-prop, we may show that this update is $\hat{\mathbf{e}}_i p$-local.

**Theorem E.13.** *If $p(\mathbf{\Theta}) = \prod_k p(\mathbf{\Theta}_k)$ and $p(\mathbf{X}|\mathbf{\Theta})$ is defined by Eq. E.39 (with $\mathbf{X} = \mathbf{r}$ and $\mathbf{\Theta} = \mathbf{W}$), the RFLO update for $\mathbf{W}_{ij}$ with a loss $\mathcal{L}(\mathbf{r}(T))$, given by $\mathcal{A}_{RFLO}(p(\mathbf{r}|\mathbf{W}), \mathcal{L}(\mathbf{r}))$ is $\hat{\mathbf{e}}_i p$-local, where $\mathbf{e}_i = \frac{\partial L(\mathbf{r}(T))}{\partial \bar{\mathbf{r}}_i(T)}$.*

## E.14    Feedback-based Online Local Learning Of Weights (FOLLOW)

The FOLLOW algorithm [50] is defined in terms of a particular continuous-time LIF circuit with postsynaptic potential kernels. For simplicity, we will focus our analysis on a linear version of the same circuit, disregarding the dynamic postsynaptic potentials and input stimuli. Disregarding these features does not affect the $p$-locality properties of the FOLLOW algorithm, but it would certainly degrade its performance on tasks.

The network dynamics are given by:

$$\mathbf{r}(t + \Delta t) = h(\mathbf{r}(t), \mathbf{e}(t)) + \sigma \boldsymbol{\eta} \tag{E.51}$$

$$= (1 - \frac{\Delta t}{\tau})\mathbf{r}(t) + \frac{1}{\tau} \left( \mathbf{W}\mathbf{r}(t) + k\mathbf{W}^{fb}\mathbf{e}(t) \right) \Delta t + \sigma \boldsymbol{\eta}, \tag{E.52}$$

where $\mathbf{W}^{fb}$ is an $N \times N^o$ random feedback weight matrix, $k$ is a positive constant, and $\mathbf{e}(t)$ is an $N^o$-dimensional error feedback vector delivered at every timestep, with $N^o$ the number of output dimensions. Because we are in continuous time, we will assume that $\boldsymbol{\eta} \sim \mathcal{N}(0, \Delta t)$.

Similar to RTRL, we can write the probability distribution for the network as:

$$p(\mathbf{r}|\mathbf{e}, \mathbf{W}) = p(\mathbf{r}(0)) \prod_{t} \prod_{i=1}^{N} p(\mathbf{r}_i(t + \Delta t)|\mathbf{r}(t), \mathbf{e}(t), \mathbf{W}) \quad \text{(E.53)}$$

$$p(\mathbf{r}_i(t + \Delta t)|\mathbf{r}(t), \mathbf{e}(t), \mathbf{W}) \sim \mathcal{N}(h(\mathbf{r}(t), \mathbf{e}(t)), \sigma^2 \Delta t), \quad \text{(E.54)}$$

where $p(\mathbf{r}(0))$ is some initial distribution of firing rates. Further, we can assume that the distribution of errors at timestep $t + \Delta t$ has any arbitrary distribution $p(\mathbf{e}(t + \Delta t)|\mathbf{r}(t))$.

The update for weight $\mathbf{W}_{ij}$ is given by:

$$\Delta \mathbf{W}_{ij}(t) \propto \left( \mathbf{W}_{i:}^{fb} \mathbf{e}(t) \right) \mathbf{r}_j(t). \quad \text{(E.55)}$$

Therefore, only the postsynaptic error current and presynaptic input are necessary to update the weights for a given synapse in this type of network. Below, we will show that this update is $p$-local.

**Theorem E.14.** *If $p(\boldsymbol{\Theta}) = \prod_k p(\boldsymbol{\Theta}_k)$ and $p(\mathbf{X}|\boldsymbol{\Theta})$ is defined by Eq. E.53 (with $\mathbf{X} = \{\mathbf{r}, \mathbf{e}\}$ and $\boldsymbol{\Theta} = \mathbf{W}$), the FOLLOW update for $\mathbf{W}_{ij}$, given by $\mathcal{A}_{FW}(p(\mathbf{r}, \mathbf{e}|\mathbf{W}))$ is $p$-local.*

*Proof.* To see that this is true, we need only show that the variables included in $\Delta \mathbf{W}_{ij}$ are subsets of the variables included in the score function $\frac{\partial \log p(\mathbf{r}, \mathbf{e}|\mathbf{W})}{\partial \mathbf{W}_{ij}}$. These variables are permissible for $p$-local updates by Property 2.7. The score function is given by:

$$\frac{\partial \log p(\mathbf{r}, \mathbf{e}|\mathbf{W})}{\partial \mathbf{W}_{ij}} = \sum_{t} \frac{\partial \log p(\mathbf{r}(t + \Delta t)|\mathbf{r}(t), \mathbf{e}(t), \mathbf{W})}{\partial \mathbf{W}_{ij}} + \sum_{t} \frac{\partial \log p(\mathbf{e}(t + \Delta t)|\mathbf{r}(t))}{\partial \mathbf{W}_{ij}} \quad \text{(E.56)}$$

$$= \sum_{t} \frac{\partial \log p(\mathbf{r}_i(t + \Delta t)|\mathbf{r}(t), \mathbf{e}(t), \mathbf{W})}{\partial \mathbf{W}_{ij}} \quad \text{(E.57)}$$

$$= \sum_{t} \frac{\left( \mathbf{r}_i(t + \Delta t) - \mathbf{r}_i(t) + \frac{\Delta t}{\tau} \left( -\mathbf{r}_i(t) + \mathbf{W}_{i:}\mathbf{r}(t) + k\mathbf{W}_{i:}^{fb}\mathbf{e}(t) \right) \right)}{\Delta t \sigma^2} \mathbf{r}_j(t). \quad \text{(E.58)}$$

Therefore, for weight $\Delta \mathbf{W}_{ij}$, the permissible variables include: $\mathbf{r}_i(t) \ \forall t$, any $\mathbf{r}_k(t)$ such that $\mathbf{W}_{ik} \neq 0$ ($\forall t$), any $\mathbf{e}_k(t)$ such that $\mathbf{W}_{ik}^{fb} \neq 0$ ($\forall t$), and the parameters $\mathbf{W}_{i:}$ and $\mathbf{W}_{i:}^{fb}$. The parameter update requires only $\mathbf{r}_j(t)$ and $\mathbf{W}_{i:}^{fb}\mathbf{e}(t)$, which is a subset of these permissible variables. Therefore, the update is $p$-local. $\qquad \square$

## F $\quad p$-locality does not guarantee biological plausibility

It is very important to clarify the exact relationship between $p$-locality and biological plausibility. Except for some network-wide variables that a theoretician may decide to allow through a particular choice of $\mathbf{S}p$-locality, we have generally shown that $p$-locality is *overly permissive*, in that a particular choice of $p$ may allow parameter updates to include variables that an individual synapse may not have access to. Furthermore, $p$-locality does *not* restrict the network architecture defined by $p$ to be biologically plausible. The best way to interpret $p$-locality is as follows: if $p(\mathbf{X}, \boldsymbol{\Theta})$ defines a biologically plausible architecture *and* an algorithm $\mathcal{A}$ is $p(\mathbf{X}, \boldsymbol{\Theta})$-local, then the parameter update provided by $\mathcal{A}(p(\mathbf{X}, \boldsymbol{\Theta}))$ will be biologically plausible. There are many network architecture and parameter update combinations that may be biologically plausible without being proven $p$-local (e.g. explicit approximations to backpropagation [29, 30, 31]), as there are many combinations that are $p$-local without being biologically plausible. Below, we will show two important instances in which $p$-locality does not properly diagnose a combination of network architecture and parameter update as biologically implausible.

### F.1 Locality and architectural plausibility

The first example is pervasive in neural network models of the brain: networks frequently violate Dale's law, which states that neurons in a neural network are (for the most part [51]) either excitatory

(outgoing weights are positive) or inhibitory (outgoing weights are negative), but not both. In fact, in the simple network example we have provided (Section 2.4), neural firing rates are not constrained to be strictly positive, and outgoing synaptic weights are not sign-constrained. For this biologically implausible architecture, $p$-locality defines which variables are allowed to be included in individual parameter updates in a way that is sensible (allowing only variables involving the postsynaptic firing rate and the firing rates of all pre-synaptic neurons), but it says nothing about the aforementioned implausibilities of the network architecture. Similarly, the linear Boltzmann machine example provided in Appendix D does not constrain firing rates to be positive, and requires symmetric weights ($\mathbf{W}_{ij} = \mathbf{W}_{ji}$), which could not satisfy Dale's law while allowing connections between an inhibitory neuron $i$ and an excitatory neuron $j$ ($\mathbf{W}_{ij} > 0$ while $\mathbf{W}_{ji} < 0 \Rightarrow \mathbf{W}_{ij} \neq \mathbf{W}_{ji}$).

These examples illustrate an important fact: $p$-locality focuses on the plausibility of updates *given* an architecture that has been predetermined to be acceptable. However, it is worth noting that if we were to impose these additional constraints for the proposed networks, the accepted variables determined by $p$-locality would not change.

### F.2   Parameterizing probabilities with neural networks

Another important caveat when working with $p$-locality is that the random variables $\mathbf{Z}$ have to correspond to the relevant biophysical quantities of interest, e.g. neural firing rates $\mathbf{X}$ and synaptic weights $\boldsymbol{\Theta}$. If this is not the case, then $p$-locality can easily defy standard notions of biological plausibility. For instance, if we define a probability distribution in terms of a 3-layer neural network:

$$p(\mathbf{X}|\boldsymbol{\Theta}) \sim \mathcal{N}(\bar{\mathbf{r}}^2(\bar{\mathbf{r}}^1(\bar{\mathbf{r}}^0(\boldsymbol{\Theta}))), \sigma^2), \tag{F.1}$$

Then the score function of this distribution is given by:

$$\frac{\partial p(\mathbf{X}|\boldsymbol{\Theta})}{\partial \boldsymbol{\Theta}} = \frac{\mathrm{d}\log p(\mathbf{X}|\boldsymbol{\Theta})}{\mathrm{d}\bar{\mathbf{r}}^2} \frac{\mathrm{d}\bar{\mathbf{r}}^{(2)}}{\mathrm{d}\bar{\mathbf{r}}^{(1)}} \frac{\mathrm{d}\bar{\mathbf{r}}^{(1)}}{\mathrm{d}\bar{\mathbf{r}}^{(0)}} \frac{\mathrm{d}\bar{\mathbf{r}}^{(0)}}{\mathrm{d}\boldsymbol{\Theta}}. \tag{F.2}$$

This equation depends on $\mathbf{X}$, which is the output of the network, even though $\boldsymbol{\Theta}$ parameterizes $\bar{\mathbf{r}}^{(0)}$. Therefore, if the random variables had been defined in as in Section 2.4, then this update would not be $p$-local. However, because it is the derivative of the score function, for any independent marginal $p(\boldsymbol{\Theta})$, it is $p$-local for this choice of random variables. Therefore, it is important when working with an algorithm such as Wake-Sleep or REINFORCE, that one chooses a conditional probability distribution $p(\mathbf{X}|\boldsymbol{\Theta})$ that captures biologically plausible dependencies. When this is not done, as in [52, 53], the resulting updates have no correspondence to synaptic plasticity rules.

Note that this fact does not undermine the utility of $p$-locality as a concept. Our proofs for algorithms in Appendix E apply for *any* $p(\mathbf{X}|\boldsymbol{\Theta})$, as long as $p(\boldsymbol{\Theta})$ factorizes to $\prod_i p(\boldsymbol{\Theta}_i)$. Therefore, algorithms that have universal $p$-local properties will respect the variable dependencies implied by $p(\mathbf{X}|\boldsymbol{\Theta})$ whether this distribution is plausible or not, which means that the algorithms will respect variable dependencies for *all* plausible network architectures.

