# OpenReview forum: "Formalizing locality for normative synaptic plasticity models"
_NeurIPS.cc/2023/Conference — NeurIPS 2023 poster_

### Official Review · Reviewer_GWia · 2023-06-19

**Soundness:** 4 excellent
**Presentation:** 3 good
**Contribution:** 3 good
**Rating:** 7
**Confidence:** 3

**Summary:**

Many biologically plausible learning algorithms have been developed in the recent years but much less effort have been deployed to compare them and understand how they could yield experimentally testable hypothesis. The present work introduces a framework that allows comparing existing algorithms. In particular, the authors introduce the concept of Sp-locality which highlights the quantities needed to be available to synaptic plasticity, the p part focusing on quantities made available by neural computations and the S one on additional signals learning signals. Different groups of plasticity rules arise from the proposed analysis, helping formulating empirically falsifiable hypotheses.


**Score increased from 6 to 7 after the rebuttal**.

**Strengths:**

The paper is technically sound, introducing formal definitions of locality and thoroughly examining their implications. It is well written, with examples and illustrative figures facilitating reader comprehension of the newly introduced concepts. The analysis conducted allows a straightforward comparison of various learning rules. Furthermore, the authors transparently outline the framework's limitations.

Those points make the paper valuable to the NeurIPS community, particularly due to the lack of prior work on the question it addresses.

**Weaknesses:**

Some of the properties deriving from Sp-locality do not seem particularly aligned with what one could picture of biological plausibility. As this may result from a misunderstanding from my side, I address those points in the question section. I am happy to improve my score once those questions are addressed.

**Questions:**

**Note:** all the following questions are based on my current (incomplete) understanding of the paper.

In a feedforward neural network, a plasticity rule for $W^l_{ij}$ (synaptic weights between layer $l-1$ and $l$) that depend on $r^{l-1}$ (activity of the **entire** $l-1$ layer) and of $r^{l}_j$ is consider $p$-local. Is this compatible with any notion of biological plausibility? This point makes me wonder whether the notion of $p$-locality is fully aligned with locality in physical space.

What are the main differences of the proposed approach with an alternative one focusing on the computational graph of neural computation? Such an approach would, for example, not suffer from the FF issue described above. It would start from the neural computations performed and end up with the learning rules. Of course, one downside is that it now depends on the details of neural computation.

How would the Sp-locality framework extend to algorithms attacking the temporal credit assignment problem, e.g. e-prop of Bellec et al. 2020 or FOLLOW of Gilra and Gerstner 2017?

Finally, which conclusions the framework enables establishing that a more vanilla approach that just analyze the learning rules “manually” does not achieve?

**Limitations:**

Yes

---

> ### Author Rebuttal · Authors · 2023-08-07
>
> Thank you for your feedback, here we will do our best to clarify the points that you raised. In some cases we will refer to the General Comments [GXX] if your question was also raised by other reviewers.
>
> Regarding your question about whether or not p-locality is fully aligned with our intuitions about locality in space: in our example from Section 2.4, we do see that updates in a MLP network are dependent on all neurons in the preceding layer. However, this is because MLP networks are all-to-all connected. If a neuron in layer $l-1$ were not directly synapsing onto & communicating with neuron $r_i$, then no synaptic updates would be able to depend on it under p-locality. We do see that some update rules depend on ALL presynaptic neurons however, rather than only the particular synapse’s presynaptic input, $r_j$. This is acceptable, however, because in this example all neuronal inputs are being summed to contribute to the membrane potential $V_i$. Within neuroscience it is well-known that synapses onto the same neuron can interact and share information (i.e. heterosynaptic plasticity, a well documented phenomenon), and if the update depends on the sum of all activity in particular, then it actually coincides quite well with our basic intuitions about spatial locality in neurons (since the sum of activity will be reflected in back-propagating action potentials in the dendrites).
>
> However, as we do note in considerable detail in Appendix F, when unrealistic or over-simplified networks are used, p-locality does not necessarily correspond to our physical, biological notions of locality. One benefit of p-locality is that the more realistic you make your biophysical neuron model, the more realistic the p-locality constraints become (which is why the examples in Section 2.4 and Appendix D produce sets of allowed variables under p-locality that closely correspond with physical, biological intuitions).
>
> For a discussion of the relationship between our formulation and one focusing on the computational graph, please refer to response [G3].
>
> This framework certainly can (and does) account for algorithms that do temporal credit assignment. For example, models that support general DAG architectures (in particular REINFORCE and Wake-Sleep) can also be used for learning through time, and these temporal network architectures do not disrupt our proof in any way because they are subsets of general DAG models (see Fig. 1b). We will add a discussion of these matters and highlight some additional specific algorithms, including  BPTT and several of its approximations (RFLO, e-prop), plus FOLLOW (Gilra & Gerstner 2017)--the derivations for these algorithms are very similar to the one provided for backpropagation (Section 3.3 and Appendix E.9).
>
> For a discussion of the relationship between our approach and a more ‘vanilla,’ manual approach, please refer to response [G4].

---

> > ### Comment · Reviewer_GWia · 2023-08-10
> >
> > Thanks to the authors for their detailed answers. Most of my questions are answered. The one about the feedforward example is not yet totally solved.
> >
> > I understand the authors’ argument as follows. Synaptic rules of the form pre-synaptic x post-synaptic values are a specific example of Sp-local rules for this kind of model but Sp-local rules are not limited to that, heterosynaptic plasticity being such an example. Is that correct? Given that this prediction is different from what a connection / computational graph approach would give, I encourage the authors to discuss this point in detail in the next version of their paper.
> >
> > Given that the authors answered my question, I am increasing my score to 7. The paper is definitely technically very strong and provides non-intuitive answers to an important question.

---

> > > ### Author Response · Authors · 2023-08-10
> > > **Reply to Reviewer GWia**
> > >
> > > Thank you for the increase in score! Your understanding is correct, that Sp-local rules are more general than just pre-synaptic x post-synaptic updates, with heterosynaptic plasticity being an important example. We will make sure to add a discussion of this to our revised manuscript.

---

### Official Review · Reviewer_p6bX · 2023-07-06

**Soundness:** 3 good
**Presentation:** 2 fair
**Contribution:** 3 good
**Rating:** 6
**Confidence:** 4

**Summary:**

The paper describes a systematic and mathematically formal method for categorizing learning algorithms into different types and degres of locality. It also applies the method to certain such algorithms and determines their type of locality.

**Strengths:**

The authors chose to work on a topic that is indeed very important, and in a field has been very much in need of systematic evaluation. The claims of biological plausibility and locality in the recent literature have been increasing and widening, so a method to validate such claims has been necessary.

The fact that the paper does not merely provide a verbal definition of types of plasticity but rather it attempts to make it mathematical is very interesting.

The framework is mostly reasonable, judging by its results when applied to the categorization of specific learning rules.

**Weaknesses:**

The weakest aspect of the framework presented here is that it falls victim to one of the issues that it intends to resolve.

More specifically, algorithms such as equilibrium propagation or predictive coding require that the network reach a global equilibrium through multiple forward and backwkard propagations before a training example completes its updating action on the network parameters. This is arguably less local than even standard backpropagation, as it requires not only that global signals propagate through the network once, but indeed multiple times. This has been recognized in recent works (https://iopscience.iop.org/article/10.1088/2634-4386/aca710, https://openreview.net/forum?id=8gd4M-_Rj1) and this recognition has even led to a formal proof that the time-complexity of these algorithms is lower-bounded by that of backpropagation (https://arxiv.org/abs/2304.02658). Therefore, this type of locality is clearly not the same as that of, for example, Contrastive Divergence for Boltzmann machines or Deep Belief Networks which do not presuppose such a global and multilayer equilibrium. In turn, even in the cases where such an equilibrium requires iteration only within individual layers and where greedy layer-wise training is possible, e.g. in Boltzmann machines and contrastive Hebbian rules, the final weight update cannot be said to be equaly local to that of simpler Hebbian rules that do not require such iterative processes or feedback, and complete the update truly locally in a single timestep. The distinction of these categories of locality is crucial. Disregarding these in a paper that claims to resolve such confusions risks reinforcing and perpetuating them instead.

The above issue of the paper might have been caused by the fact that many of the simplest and most local learning rules are not normative but rather are based on heuristics, whereas the presented framework is concerned with normative rules. However, there do exist such rules that are normative, eg the type of STDP in Nessler et al., NIPS 2009/Plos Comp. Biol. 2013; the work of Pehlevan & Chklovskii, NIPS 2015; and SoftHebb in Moraitis et al., Neuromorph. Comput. Eng. 2022. In fact, what appears to be the current state of the art in biological plausibility of deep learning and in performance of bio-plausible deep learning (Journé et al., ICLR 2023) is one of the above normative rules, namely SoftHebb.

Therefore, I recommend that the authors adjust their framework to account for the substantial differences in locality between the three categories of algorithms that are exemplified by (a) Equilibrium Propagation, (b) Contrastive Divergence/Boltzmann Machine, and (c) SoftHebb or STDP. This should be accompanied by making sure that examples of all these categories are included in Table 1.

As a minor note, the paper's Table 1 mentions Boltzmann Machines as an algorithm, but BMs are models. I believe Contrastive Divergence is meant by the authors, and if so it should be corrected.

More generally, the fact that the current formalization suffers from similar issues as other less formal claims in the literature, reveals that perhaps the mathematics in this case are superfluous, since its conclusions depend mainly on the assumptions, which in turn depend on the categories that the authors intend to achieve. In other words, I suspect that the main benefit of a systematic categorization could be achieved without mathematical terms. Could the authors please comment on this? What do they regard as added value from the use of mathematics instead of mere natural language and logical arguments in this case?

Related to this, the paper would be strengthened significantly if both the framework and the resulting categories of locality could be described in a dedicated paragraph in natural language, with an intuitive explanation.

**Questions:**

"We subsequently use this framework to distill testable predictions from various classes of biologically plausible synaptic plasticity models that are robust to arbitrary choices about neural network architecture"
This is a quote from the abstract. Could the authors please elaborate on this? What exactly are the testable predictions? Perhaps this refers to the last paragraph of the paper, but that paragraph is not entirely clear to me, and it also seems like it does not make any specific prediction.

If the authors could address these issues both in the revised manuscript and in their rebuttal here, I am open to re-evaluating the submission.

**Limitations:**

There is no discussion of the paper's limitations in the paper.

---

> ### Author Rebuttal · Authors · 2023-08-07
>
> We thank the reviewer for their very detailed comments! We will respond to your specific comments here, and for questions asked by multiple reviewers, we will refer you to the appropriate response in the General Comments section [GXX]. All of our comments below will be incorporated into our subsequent draft.
>
> We agree with the reviewer that a good framework for analyzing locality should contend with the difference between algorithms that require MCMC sampling and those that do not. If, as the reviewer suggests, our framework masked this distinction that would indeed be problematic. However, that is not the case. Indeed, our definition of Sp-locality requires one to determine the architecture required for locality, and some architectures will not require MCMC (such as a DAG), while others will (such as an energy-based model). Thus, Sp-locality does not ignore the question of whether MCMC sampling is required or not. We will be sure to clarify this in our revisions.
>
> However, a related though slightly different question that the reviewer raises is how many samples of MCMC are required by an algorithm for effective training. Here, we agree with the reviewer that our framework does not fully account for this. But, it does partially. Indeed, we should clarify the similarities and differences between Contrastive Divergence, Equilibrium Propagation, and more typical Hebbian rules in this respect. In particular, we note that the Nessler et al. NIPS 2009/Plos Comp. Bio 2013 papers construct their learning algorithm as a very close approximation of the generalized Expectation Maximization (GEM) algorithm, which is as a consequence $p_m$-local (see Table 1), where $p_m$ is defined by the generative model used in these papers. While generative models typically have neither analytically tractable nor biologically plausible inference distributions, by making a careful choice of generative model, these authors are able to perform exact inference through a type of winner-take-all neural circuit. As a consequence, they do not need to rely on slow MCMC dynamics or MAP inference, and are also able to avoid offline ‘sleep’ phase sampling (this is why this model does not require information about a global clamping variable $\gamma$). This sets this family of algorithms apart from both Contrastive Divergence and Equilibrium Propagation, and our framework captures this. Similarly, SoftHebb derives its locality via very similar mechanisms without requiring sampling from multinomial variables, and again, this can be identified via our formalisms. We will add these models to Table 1, and will note their relationship to the GEM algorithm in the appendices.
>
> But, in line with the reviewer’s point, it is true that Contrastive Divergence does not require full mixing for MCMC sampling, unlike Equilibrium Propagation, and yet this fact does not affect the p-locality of the Contrastive Divergence learning rule versus Equilibrium Propagation or Predictive Coding (though it does affect the supported architectures for the learning rule, see column 2, Table 1). This raises an interesting possibility: would it be possible to extend our framework to not only distinguish models that require MCMC sampling from those that don’t, but also to distinguish models that require only a little bit of MCMC sampling from those that require a lot? We think this is a fascinating potential direction for future work building on our framework here. We will add a note on limitations highlighting this point to our paper, and emphasize that future work should address this distinction. To ensure that we are not adding to any confusion in the literature we will also add a note in column 2 Table 1 to emphasize that Contrastive Divergence is somewhat more flexible than predictive coding and equilibrium propagation in terms of its time complexity, because it requires only K-Step MCMC sampling where K is a small integer.
>
> Further, thank you for catching our misuse of terminology. We will change our use of ‘Boltzmann Machine’ to ‘Contrastive Divergence’ in Table 1 and in the appendices.
>
> For a discussion of the relationship between our approach and a more manual/linguistic approach, please refer to response [G4].
>
> We provide a detailed explanation of the different Sp-locality categories, as well as their experimental predictions in General Comments [G2]. For a broad description of the approach in general, we refer to Lines 45-59 of the main text. We will use these to expand and clarify the final paragraph in the Discussion section.

---

> > ### Comment · Reviewer_p6bX · 2023-08-19
> > **Remaining concern**
> >
> > I would like to thank the authors for their clarifications.
> > These are helpful, and the promised changes would bring important improvements.
> >
> > However, my remaining concern is an essential unresolved part of my main concern from my earlier comment. It is the following. I believe that the "p_m-local" category is too broad if it considers as equally local e.g. (a) simple hebbian learning over a single layer and in a single timestep, and (b) predictive coding that involves propagating over multiple layers and in many timesteps for each learning example. Clearly, learning of type (a) is much more local than learning of type (b). The framework should cover this difference, but it does not, as it groups both very local and very non-local models in "p_m".
> >
> > Again, the reason why I focus on this is because I anticipate that this will continue perpetuating such severe misunderstandings in the literature.
> >
> > I presume that the authors might not be able to formalize this in the time remaining for this paper. In that case, what are the other exact changes that the authors will make to the manuscript to emphasize the existence of these important differences in locality?

---

> > > ### Author Response · Authors · 2023-08-20
> > > **Response to additional concerns**
> > >
> > > Thank you for your response! Though predictive coding and SoftHebb learning do require similar information for their parameter updates, and are consequently both $p_m$ local, you are correct that the network architectures themselves used in these two different algorithms are very different (predictive coding requires the computation of a dynamic equilibrium that takes many time steps, whereas SoftHebb and similar rules require a single time step in a winner-take-all circuit). The way that we have handled these differences in our formalism is to note in the second column of Table 1 the network architectures required for an algorithm to satisfy a given $Sp$-locality constraint. To indicate the various implausible aspects of predictive coding, we have noted that the algorithm requires MAP estimation through equilibrium dynamics, and have also indicated that the algorithm requires weight symmetry. For SoftHebb and its variants, we will specify that the $p_m$-locality has been proven for a winner-take-all circuit. Furthermore, we will provide more details about these circuits in the supplemental material where we prove the networks’ $Sp$-locality properties. We will also take great care to emphasize in our discussion that $Sp$-locality can only be informative when one also considers the types of network architectures supported by a given algorithm (in this case MAP estimation vs. winner-take-all dynamics). This will be added in addition to the limitation mentioned in our previous comment, that our current framework does not distinguish between the amount of time required for MCMC sampling across different algorithms.

---

> > > > ### Comment · Reviewer_p6bX · 2023-08-20
> > > >
> > > > As I mentioned in the previous comment, fully correcting this issue would involve formalizing these separate types and degrees of locality as such, instead of grouping them in the same type of locality.
> > > >
> > > > Nevertheless, the authors' suggested changes do bring important improvement that alleviate my most significant concern partly. I am accordingly raising my score.

---

### Official Review · Reviewer_X8HR · 2023-07-06

**Soundness:** 3 good
**Presentation:** 3 good
**Contribution:** 3 good
**Rating:** 7
**Confidence:** 3

**Summary:**

The paper introduces a general definition of locality of updates in neural networks, which is an important requirement of biological plausibility. Variants of locality requirements allow to describe classes of networks and algorithms. Several important algorithms are analyzed through this lens.

**Strengths:**

Definition of locality (and variants) is novel (to my knowledge), considerations are mathematically sound.

The paper outlines several patterns of locality in existing algorithms, which deserves interest.

**Weaknesses:**

The need for the general definition of locality is not substantiated sufficiently. Was that need already expressed in the literature? Were there other attempts? Is there an example where such definition would resolve a controversy?
Whenever a model is formally defined, its biological plausibility (which of course goes beyond locality) is up for debate. However within the model the information available for parameter updates must be clear from the definition (or I can't think how it can not be). Therefore I am not convinced that suggested framework will help "guide claims of biological plausibility", as the paper says.

It is not quite convincing that introducing probabilities is worth the trouble. After all, the answer to locality question is always binary: yes or no. It seems that definition through graphs (directed or undirected) would be sufficient in overwhelming majority of cases, and whenever the probabilities are already defined, the graphical model can be referred to.


**Questions:**

The authors claim there is lack of clarity about locality for some models. Can you give an example?

Definition 2.1: line 87, what are you trying to say by underscoring the word 'direct'? what would be indirect?

Fig. 1b Node r_j for t_2 is not a child or coparent of W_{ij} but not excluded.

Coloring in figures can be guessed, but better be explained in words.

Eq.4 How would the definition 2.2 work when $h$ is not differentiable, e.g. ReLU?

line 239 A(p(X, \Theta)) = f(Z) - it looks like update function is somehow fully determined by the probability distribution? Please define the meaning of A(p(...)) here and in theorems 3.1, 3.2.


**Limitations:**

No problem nere

---

> ### Author Rebuttal · Authors · 2023-08-07
>
> First, we thank the reviewer for their detailed feedback on our manuscript. We will make sure to add relevant material in relation to our responses below in our subsequent draft.
>
> For a discussion of the current state of the field, and an explanation of what our framework adds to discussion surrounding experimental validation of existing models, please refer to our general reviewer responses [G1] and [G2]. But, to expand on these points a little bit here, there are two key reasons that these formal mechanisms were needed. First, in contradiction to the reviewer’s intuition, it is not the case that the information required for model updates is always apparent based simply on the definition of the learning rule. Consider, for example, the predictive coding algorithm. The update rules themselves appear to be purely local, and at first blush, they provide the impression of clear biological plausibility. Yet, when analyzed under the lens of Sp-locality, we see that these apparently local equations depend on an architectural assumption of maximum a posteriori gradient descent dynamics, which is biologically questionable. Thus, one benefit of Sp-locality is precisely that it forces us to actually contend with the assumptions being made to achieve locality. Without this formal tool it can be easy for researchers to convince themselves that something is “biologically plausible” because it is local, while ignoring the assumptions required to achieve that locality. Second, our framework based on probabilities is useful beyond a basic verbal description because it provides a formalized and standardized method for describing the biological plausibility of a given plasticity model–beyond this, it also abstracts away details that are not useful for comparison across different network architectures and algorithms (see [G2]).
>
> For a full discussion of our particular choice to use probability distributions to ground our definition of locality, please see [G3]. Here we will specifically justify our choice to use stochastic—rather than deterministic—computation graphs. More than half of the algorithms listed in Table 1 were originally derived in terms of probabilistic networks. Given that the probabilistic formulation is more general (we can always look at no-noise asymptotes for the deterministic case), it makes sense to work in probabilistic terms. Further, the probability distribution provides the cleanest mechanism for deciding how to structure the edges and nodes of the computation graph, as we describe in [G3].
>
> Fig. 1b Node $r_j$ for $t_2$ is not a child or coparent of $W_{ij}$ but not excluded. This is a good catch, thank you! It is a coparent for $W_{ij}$ at the next time step--we'll add a line that is implicitly connecting to the next time step to clarify this. We will also explain the color coding in more detail.
>
> We only need our update f in Eq. 2 to be differentiable at a single point and for that derivative to be nonzero, so for a ReLU this would be fine (it's differentiable and nonzero for all x > 0). In this case our assumption of differentiability is not so important. The Heaviside function is a better example, because at the only point at which it does depend on a variable, the derivative is not well-defined. In this case, there is no clear way for the derivative operation to detect a functional dependence. A way to relax our assumption of differentiability would be to require that there are two values of $Z_i$ such that $f(Z_{i1}) \neq f(Z_{i2})$ on the left side. This would not disrupt anything for S-locality, but would potentially disrupt our proofs for the properties of p-locality, so we chose to require differentiability throughout. Another alternative would be to use a function that very closely approximates the heaviside function but preserves its differentiability (e.g. sigmoid(k x), where k >> 1).
>
> Clarifying the meaning of $\mathcal A(p(X, \Theta)) = f(Z)$. The notation here indicates that an algorithm $\mathcal A$ receives a probabilistic network architecture $p(X, \Theta)$, and outputs a parameter update function $f(Z)$. This update function changes if the probability distribution changes. For instance, the REINFORCE update Eq. 6 changes if we change our choice of nonlinearity $h$ which is used to parameterize our probability distribution over network states. One of our highly nontrivial results is that we can show the REINFORCE update will be Rp-local no matter what choice of probability distribution p we make.

---

> > ### Comment · Reviewer_X8HR · 2023-08-14
> >
> > Most of my concerns have been addressed, thank you.
> > I think the work should be made available to the community, where the discussion will most likely continue. The score upped to 7.

---

### Official Review · Reviewer_1DnT · 2023-07-25

**Soundness:** 4 excellent
**Presentation:** 4 excellent
**Contribution:** 2 fair
**Rating:** 6
**Confidence:** 3

**Summary:**

The authors propose "Sp-locality" as a formal definition of locality in models of synaptic plasticity, as well as two intermediate definitions, S-locality and p-locality. S-locality explicitly enumerates the set of variables which directly participate in a synaptic update for each synapse. p-locality replaces the explicit set S with the Fisher information of that variable (conditioned on all other variables to eliminate indirect dependencies) with respect to the synapse. Finally, Sp-locality expands this by measuring influence as defined by p-locality, additionally allowing a dependence of the synaptic update on an explicit set of variables as with S-locality (although eliminating the need for an exhaustive enumeration). Finally, the authors apply these definitions to a number of well-known models of synaptic plasticity, delineating four major classes of models and suggesting experimental predictions.

**Strengths:**

I am unaware of similar work attempting to organize plasticity rules according to their locality, although I am interested to see what other taxonomies have been suggested. The paper is exceptionally well written, building up the concept of Sp-locality from simpler intuitive concepts, providing examples and intuitions, and expanding on these in the Appendices. The authors classify a wide range of models using their framework and show useful mappings between the graphical model and their definitions of locality. Overall, this paper is a useful initial effort to formally categorize "local" plasticity rules, and is a valuable conceptual contribution and starting point for follow-up work in this direction.



**Weaknesses:**

- Although the authors do a thorough job of applying their framework to a diversity of plasticity models, I am curious about further contextualization of the work within meta-topic of "taxonomy of plasticity rules"
- As the authors point out, p-locality does not imply biological plausibility (nor vice versa). Sp-locality is an even weaker constraint, as it allows for not only p-locality but also dependence on an explicit subset S of network variables. These definitions punt the problem of "biologically plausible plasticity" to the problem of defining "biologically plausible architecture," which is again often defined intuitively and ad-hoc. As such, the practical utility of this framework is unclear to me.
- It is unclear to me how this framework adds power to experimental predictions beyond the ones made by the individual plasticity rules and their ad-hoc appeals to biological plausibility. I would be interested to see more discussion about this with explicit examples of predictions which can be made with this framework and followup experiments that can be performed (even conceptually, disregarding their practical feasibility), or even a retrospective "post-diction" of a known experimental phenomenon which could have been predicted by this framework.


**Questions:**

- Is there other work attempting a conceptually similar grouping of plasticity rules based on the information used by synapses?
- Can these definitions account for temporal locality or lack thereof? e.g. how would backpropagation through time or its biologically plausible approximations fit into this framework?
- Is there a mapping between p-locality and S-locality? i.e. given a p-local rule (for a particular architecture), is there always a way of defining S-locality, and vice versa? (Is this even a useful mapping to make?)
- Are there plasticity models that do not fit into this framework (excepting the trivial cases in which S=Z)?


**Limitations:**

The authors carefully analyzed the limitations of their definitions, although as far as I can tell these limitations severely handicap the practical utility of the proposed framework. Due to the basic science nature of this work, potential for direct societal impact is minimal.

---

> ### Author Rebuttal · Authors · 2023-08-07
>
> Thank you for your very comprehensive review; we will elaborate here on your criticisms and requests for clarification. All of our comments below will be incorporated into our subsequent draft.
>
> For our response on previous related work, please see general comment [G1].
>
> It is a genuine issue that our framework does not answer the question: “which network architectures are biologically plausible?”. Whether a mathematical framework could (or should) be constructed that would answer this question is unclear; arguably, that question can only be answered by careful anatomical and neurophysiological investigations. However, it's worth noting that Sp-locality does not completely punt the question of “biological plausibility” to that of “biologically plausible architecture”. A proof of Sp-locality necessarily comes with the class of models that the locality properties have been proven for (see Table 1). This helps us to make clear the generality & restrictions of various local learning algorithms and facilitate their comparison. In the ideal case, an algorithm would be Sp-local for any network architecture, because this would guarantee that an algorithm at least does not restrict a modeler to using networks that violate important biological plausibility assumptions. Of course, if an algorithm is only Sp-local for a given type of architecture then one is still left with the question, “is that type of architecture biologically plausible?”. But, critically, Sp-locality forces us to contend with this fact: that is its practical utility. Without it, researchers can (and often do) mask the biologically implausible architecture assumptions they are making when they claim that their algorithm is biologically plausible because it is “local”. Put another way, the utility of Sp-locality here is to eliminate any uncertainty as to the architectural assumptions being implicitly made in claims of locality.
>
> For our response on experimental predictions made by our framework, please refer to general comment [G2].
>
> This framework certainly can (and does) account for algorithms that do temporal credit assignment. For example, models that support general DAG architectures (in particular REINFORCE and Wake-Sleep) can also be used for learning through time, and these temporal network architectures do not disrupt our proof in any way because they are subsets of general DAG models (see Fig. 1b). We will add a discussion of these matters and highlight some additional specific algorithms, including  BPTT and several of its approximations (RFLO, e-prop), plus FOLLOW (Gilra & Gerstner 2017)--the derivations for these algorithms are very similar to the one provided for backpropagation (Section 3.3 and Appendix E.9).
>
> There is always a mapping from a p-local rule to an S-local rule for a given network architecture–to see that this is true, we need only recognize that for every p-local rule, there is a finite set of allowed variables for each parameter. We may take $S_k$ for parameter $k$ to be equal to this set, thus demonstrating that the rule is also S-local. Of course, S-locality is in general radically less concise than p-locality, because without the machinery p-locality provides, it is not possible to automatically and cleanly generate a set of allowed variables, which would otherwise have to be done manually. It is also radically less useful because the set S is architecture-specific, and cannot be used for proving general properties of learning algorithms as we do in Table 1. By contrast, there is not always a mapping from an S-local rule to a p-local rule. This is because p-locality itself makes a hard commitment to an intuitive notion of locality defined on probability distributions, and does not accommodate additional variables that violate this intuitive notion.
>
> To our knowledge, there are no normative plasticity models that do not fit into our framework. In fact, we have been unable to find normative plasticity models that do not fit into the four broad categories outlined in Table 1. However, it is worth noting that p-locality becomes less useful for non-normative (e.g. phenomenological or mechanistic) plasticity rules, where models are normally highly architecture-specific, and there is less value in constructing an architecture-general notion of locality, because in these cases there is no proposed generic optimization algorithm (e.g. REINFORCE) producing different plasticity rules for different neural network architectures.

---

> > ### Comment · Reviewer_1DnT · 2023-08-12
> >
> > Thanks to the authors for their careful responses to my questions. I have raised my score from 5 to 6 and look forward to reading the extended discussion in the final version, particularly the comments regarding biologically plausible architectures and normative plasticity models, which has helped me better understand the scope of this work.

---

### Author Rebuttal · Authors · 2023-08-07

#### General comment

We would like to thank the reviewers for their extremely helpful feedback. Here, in the global reply, we will address critiques that were raised by multiple reviewers. Importantly, we are grateful that most reviewers recognized the importance of a formal framework to study learning algorithm locality. We believe our work is an important first step in that direction, and that it has the potential to be used widely. We also note that our manuscript is considerably improved thanks to excellent reviewer feedback. As such, we believe it is timely, and that it is worthy of sharing with the NeurIPS community.

[G1] Previous work.
First, in our introduction we will cite other papers that have attempted to construct taxonomies of locality in the past and clarify our unique contributions. To our knowledge, no previous study has attempted to construct a mathematical framework for locality, but some papers have reviewed locality as a whole and discussed potential ways to test differences between different proposed normative plasticity rules. In particular, the four p-locality categories that we uncover with our analysis loosely correspond to those identified in Lillicrap et al. 2020. But, our mathematical framework provides additional useful distinctions, e.g. it identifies target-based, generative learning algorithms like Wake-Sleep as distinct from error-based learning algorithms derived from backpropagation, because these algorithms derive their locality from very different principles and make different assumptions about the nature of feedback. In addition, Marschall et. al 2020 complements our approach by discussing constraints on the allowed computations, memory, and temporal complexity of bio-plausible temporal credit assignment algorithms and taxonomizes existing approaches accordingly. These amount to constraints on the network architecture itself, and constraints on the types of operations allowed in computing updates (e.g. matrix-vector multiplication), whereas we focus on allowable variables for updates, agnostic to the update’s functional form.

[G2] Clarifying experimental predictions.
We discuss this point briefly on Lines 351-55, but will elaborate here (and in our revised discussion).  Critically, Sp-locality abstracts away details that are not important for testing predictions and helps identify those that are. For example, suppose we are working with the feedforward network model of Section 2.4. Sp-locality tells us that the REINFORCE (Eq. 6) and Wake-Sleep algorithms (Eq. 8) are equivalent with respect to their ‘p’ in p-locality. This tells us that the set of allowed local variables under p-locality (e.g. pre- and post-synaptic information) are not the best targets for experimental testing between these algorithms, and instead the focus must be on the variables included in ‘S’.
Turning to this, we have identified four different kinds of Sp-locality, which vary based on the choice of set S (S = the reward signal $R$, the empty set, the global clamping variable $\gamma$, or the error feedback signal $e_i^{l}$). REINFORCE is of the first kind, and thus predicts that a global reward signal will modify synaptic plasticity. In contrast, Wake-Sleep is of the third kind, and thus predicts that there is a  global clamping variable $\gamma$ that switches the network between different modes of synaptic plasticity independently of reward. Therefore, across these different Sp-locality classes we have very different prescriptions for what to look for experimentally in terms of signals that regulate synaptic plasticity globally.

[G3] Probabilistic versus Graphical locality.
Several reviewers have suggested that we might alternatively formulate locality in terms of the graph of computations performed by a given network, rather than in terms of a probability distribution over network states. However, within each probabilistic node in our formulation there are several computations being performed (linear matrix + pointwise nonlinearity in the MLP case). A graphical approach would have to arbitrate which collection of computations counts as a ‘node’ on the graph. There are several different ways to do this: one way is equivalent to our approach, where the graph is taken to be a minimal DAG or UG which characterizes the distribution p. We demonstrate that this graph can be used to read out the allowed variables under p-locality (Properties 2.1 and 2.2), and we find that the reviewers are certainly correct that these properties are easier to use than the basic definition of p-locality itself. However, the definition of p-locality is still required to justify these properties and to unify them across different graph types (DAG vs. UG). Other approaches for defining the appropriate computation graph would be problematic, primarily because they would be more arbitrary, and consequently could not be used to prove general locality properties of normative learning algorithms.

[G4] Language versus math.
With any mathematical framework, as several reviewers have noted, it is critical to ask whether  natural language would have sufficed. In our case, the principal added value of Sp-locality is that it disentangles the locality properties of an algorithm from the specific network architecture used in the model, by exploiting mathematical  insights into how algorithms actually achieve locality (Properties 2.5-2.7). This allows us to focus on properties of normative plasticity algorithms as a whole, without nitpicking details about their particular instantiations for particular model choices, something that could not have been done without our mathematical formalisms.

---

### Decision · Program_Chairs · 2023-09-21

**Decision:**

Accept (poster)

**Comment:**

Synaptic plasticity rules are essential in (computational) neuroscience to understand learning in nervous systems. Plasticity rules are often termed "biologically plausible", however, a clear concept for this classification is missing. One particularly important feature in this respect is locality of the rule in space and time. The authors propose a formal definition of different classes of locality. It is used to propose testable predictions of plasticity models.

All reviewers agree that the manuscript is very well written. The approach is novel and it is addressing a question that is highly relevant for computational neuroscience. Reviewers note that the paper is mathematically rigorous and that the framework is well-motivated. The framework is applied to a wide range of models which provides useful categorization and insights.

One concern was that a locality category may be too broad as an equivalence class, including rules of rather different locality. While a full resolution was not possible within the discussion period, the reviewer acknowledged the suggested improvements and voted for acceptance.

In summary, the manuscript proposes a very interesting framework to understand locality in plasticity rules in a more rigorous manner. Its main strength is its novelty.